# Population genomics and epigenomics of *Spirodela polyrhiza* provide insights into the evolution of facultative asexuality
Yangzi Wang [1,2,8], Pablo Duchen[1,2,8], Alexandra Chávez [1,2,3], K. Sowjanya Sree[4], Klaus J. Appenroth[5], Hai Zhao [6], Martin Höfer[1,2], Meret Huber[1,3] & Shuqing Xu [1,2,7] ✉

Many plants are facultatively asexual, balancing short-term benefits with long-term costs of asexuality. During range expansion, natural selection likely influences the genetic controls of asexuality in these organisms. However, evidence of natural selection driving asexuality is limited, and the evolutionary consequences of asexuality on the genomic and epigenomic diversity remain controversial. We analyzed population genomes and epigenomes of *Spirodela polyrhiza*, (L.) Schleid., a facultatively asexual plant that flowers rarely, revealing remarkably low genomic diversity and DNA methylation levels. Within species, demographic history and the frequency of asexual reproduction jointly determined intra-specific variations of genomic diversity and DNA methylation levels. Genome-wide scans revealed that genes associated with stress adaptations, flowering and embryogenesis were under positive selection. These data are consistent with the hypothesize that natural selection can shape the evolution of asexuality during habitat expansions, which alters genomic and epigenomic diversity levels.

Understanding the evolution of sexual reproduction has long been at the center of evolutionary biology. Theories suggest that asexual reproduction is beneficial for the short term but costly for the long term, mainly due to accumulations of deleterious mutations and low effective population size[1–5]. Facultative asexuality, where organisms can reproduce both sexually and asexually depending on environmental conditions, should be optimal for one individual's lifespan[6,7]. While rather few animals such as aphids (Aphidoidea)[8], water fleas (Cladocerans)[9], and rotifers[10] reproduce facultatively asexually, up to ~80% of the flowering plants, including important crops and keystone species, can reproduce both sexually and asexually[11]. Asexual reproduction in plants involves different types of vegetative reproduction (e.g. runners, tubers, bulbs, corms, suckers, plantlets), as well as apomixis, the formation of seeds without fertilization[12]. Because changes between sexual and asexual reproduction affect the ability to persist in the short and long term, natural selection might act on the genetic controls of sexual and asexual reproduction in facultative asexual organisms, which in turn can alter the levels of genomic diversity, heterozygosity and effectiveness of selection in the population[2,5,13–15]. However, direct evidence

supporting this prediction remains scarce, mainly due to the lack of a suitable facultative asexually reproducing system in which the signature of selection can be detected at genomic levels.

Evolutionary changes in sexual and asexual reproduction might also affect the maintenance and dynamics of chromatin marks, e.g., epigenetic markers such as DNA methylations. In plants, cytosine methylation can occur in three sequence contexts: CpG, CHG, and CHH (H = A, T, or C), which are controlled by different mechanisms and have different dynamics during reproduction[16]. Typically, CpG and CHG methylation are maintained by methyltransferases1 (*MET1*) and CHROMOMETHYLASE3 (*CMT3*), respectively, whereas CHH methylation is mostly maintained by *CMT2*[17]. During sexual reproduction, DNA methylations are highly dynamic[18]. In both male and female gametogenesis, the megaspore mother cell and microspore mother cell experience dramatic chromatin changes during cell specification, such as heterochromatin decondensation and an enlarged nuclear volume[19,20]. During male gametogenesis, sperm DNA is highly methylated in the CpG and CHG context but has low CHH methylation in retrotransposons[18,21,22]. During female gametogenesis, CpG

[1]Institute of Organismic and Molecular Evolution, University of Mainz, 55128 Mainz, Germany. [2]Institute for Evolution and Biodiversity, University of Münster, 48161 Münster, Germany. [3]Institute of Plant Biology and Biotechnology, University of Münster, 48161 Münster, Germany. [4]Department of Environmental Science, Central University of Kerala, Periya 671320, India. [5]Matthias Schleiden Institute — Plant Physiology, Friedrich Schiller University of Jena, 07743 Jena, Germany. [6]Chengdu Institute of Biology, Chinese Academy of Sciences, 6100641 Chengdu, China. [7]Institute for Quantitative and Computational Biosciences, University of Mainz, 55218 Mainz, Germany. [8]These authors contributed equally: Yangzi Wang, Pablo Duchen. ✉e-mail: shuqing.xu@uni-mainz.de

https://doi.org/10.1038/s42003-024-06266-7                                                                                                **Article**

and CHH methylation remains largely steady[23]. After fertilization, CHH methylation increases during embryogenesis and can approach 100% at individual cytosines, which then decreases likely through a passive mechanism after germination[24–26]. In contrast, during vegetative reproduction, DNA methylation is likely steady since meiosis and embryogenesis are lacking[27–29]. Although Niederhuth, C. E. et al.[30]. comparing DNA methylations among 34 angiosperm species suggested that clonally propagated species often have low CHH methylation, the extent to which asexual reproduction affects genome-wide methylation levels remains unclear.

Here, we investigated the population genome and epigenome of a facultatively asexual plant, *Spirodela polyrhiza* (the giant duckweed; Lemnaceae), using samples from a global collection. This species, like other duckweeds from the genera *Spirodela*, *Landoltia* and *Lemna*, is characterized by leaf-like fronds derived from fused stems[31] and, with multiple roots on each frond[32] and with a highly reduced vascular system[33]. *Spirodela polyrhiza* reproduces vegetatively via budding under normal conditions but very rarely switches to sexual reproduction under unfavorable conditions[34,35]. Recent studies showed that despite its global distribution in diverse habitats, the genomic diversity, spontaneous mutation rates and DNA methylation levels in *S. polyrhiza* are very low[36–41], which might be associated with its overall low frequency of sexual reproduction. DNA methylation profiling of two genotypes suggests that DNA methylation in *S. polyrhiza*, which is substantially lower than in other plants, varied between genotypes[41]. Further insights into the evolutionary origin and consequences of asexuality on genomic and epigenomic variation in *S. polyrhiza* are required to understand the demographic history and to identify the footprint of selection on the genome.

## Results

### Extremely low genomic variations in *S. polyrhiza*

We sequenced the genomes of 131 globally distributed *S. polyrhiza* genotypes with an average of ~25 X coverage. Together with previously published samples[36,37], we analyzed the genomic diversity of 228 *S. polyrhiza* individuals across five continents (Supplementary Data 1). We identified 1,241,981 high-quality biallelic single-nucleotide polymorphisms (SNPs) and 166,075 short insertions and deletions (INDELs, less than 50 bp of length). Based on an updated genome annotation of *S. polyrhiza* (see Supplementary Results Methods 1.1 and Supplementary Results Section 2.1), we found that most of the SNPs (70.3%) are in the intergenic regions (Supplementary Fig. 1). Of all the SNPs located in the protein-coding regions, 61,039 were identified as nonsynonymous and 44,287 as synonymous (Supplementary Data 2). Consistent with our previous study[36], the genome-wide nucleotide diversity is 0.0016 (Supplementary Table 1), which falls within the lower range of genome-wide nucleotide diversity of other tested multicellular eukaryotes (Supplementary Table 2 and Supplementary Fig. 2). The species-wide efficacy of selection ($\pi_N/\pi_S$ ratio) is 0.37, the highest among studied organisms[42], indicating a relatively relaxed purifying selection in *S. polyrhiza*, despite its large effective population size[36,37].

In addition to SNPs and small INDELs, we also characterize the genome-wide structural variations (SVs, ≥50 bp in length) in *S. polyrhiza* (see Supplementary Methods Section 1.2 and Supplementary Results Section 2.2). We identified 3,205 high-quality SVs, including 2,089 deletions, 291 duplications, and 825 insertions. Among all identified SVs, 155 duplications and 169 deletions affected protein-coding sequences (Supplementary Table 3 and Supplementary Data 3). Using a permutation approach at a genome-wide level, we identified gene families that are significantly enriched with SVs and small INDELs (see Supplementary Methods Section 1.3, 1.4, and 1.5, Supplementary Results Section 2.3, and Supplementary Data 4 and 5), respectively. We found several gene families related to defences, such as *RPP8*[43] and the glycoside hydrolase[44], are enriched with both SVs and small INDELs. This is consistent with findings from *Arabidopsis* and other plant species, which show that SVs are enriched in stress and pathogen resistance[45,46] (Supplementary Data 5). Interestingly, we also found SVs and small INDELs are also enriched in gene families that are involved in organ development and reproduction, such as the receptor-like

protein kinases gene family[47] and MADS-box gene family that has been shown to have substantial gene losses and copy number variations in duckweeds[48–50].

### Population structure and demographic history of *S. polyrhiza*

Because *S. polyrhiza* is facultatively asexual, genotypes collected from the geographic proximity can be derived from the same clonal family. Using a previously established grouping threshold that was developed in *S. polyrhiza*[2], we identified 159 likely clonal families in the sampled population (Supplementary Data 6).

Population structure and principal component analyses revealed four populations in the sampled *S. polyrhiza* (Fig. 1a and b). Consistent with our previous study, the four populations are largely concordant with their geographic origins, namely America, Southeast Asia (SE-Asia), Europe, and India (Supplementary Fig. 3), with a few exceptions that can be due to recent migration events or artifacts during long-term duckweed maintenance.

We inferred the population history with a Maximum Likelihood (ML) phylogeny and Approximate Bayesian Computation (ABC). For ML, we used *Colocasia esculenta* (from the Araceae family) as an outgroup. The maximum likelihood phylogeny of all 228 genotypes indicated an early split of the American population from the other populations and subsequent splits of the Indian and European populations from SE-Asia (Fig. 1c). The European population constitutes the most recent split (Fig. 1c and d). Here, genotypes collected from the transcontinental region (e.g. Russia) showed intermediate features of SE-Asian and European populations, suggesting this as a likely migration route. Furthermore, the Indian population possibly originated via Thailand and Vietnam, as genotypes from these countries show intermediate features between Indian and SE-Asian populations.

We modeled the demographic history using an ABC modeling approach to further validate the evolutionary history of the four populations in *S. polyrhiza* (see Supplementary Methods Section 1.6 and Supplementary Table 4). Based on the phylogenetic analysis, we simulated three plausible demographic scenarios, allowing for either the SE-Asian, American or an additional putative population to function as the ancestral population (Supplementary Fig. 4). We found that the scenario, in which the American population and Asian population were derived from an additional putative ancestral population, constituted the most supported model (Fig. 1d). While the American population was separated from other populations around one million generations ago, the European population was derived from the SE-Asian population only 12,000 generations ago (see Supplementary Results Section 2.4 and Supplementary Table 5).

### Determinants of genomic diversity among populations

Among the four populations, nucleotide diversity ($\pi$) and the efficacy of selection ($\pi_N/\pi_S$ ratio) varied among populations (Fig. 2b). While the SE-Asian population has the highest $\pi$ and lowest $\pi_N/\pi_S$ ratio, the American population has the lowest $\pi$ and highest $\pi_N/\pi_S$ ratio. Interestingly, while the European population has a much smaller $\pi$ compared to the SE-Asian population, the $\pi_N/\pi_S$ ratio of the European population remains similar to the latter, likely due to its recent split from the SE-Asian population.

Using genome-wide SNPs, we found that linkage disequilibrium (LD) is comparable to *Arabidopsis thaliana*[51], suggesting considerable historical sexual reproduction in *S. polyrhiza*. However, the extent of LD decay varied substantially among populations (Fig. 2b and Supplementary Fig. 5). While the Asian population showed the most rapid LD decay (about 12 kb at $r^2 = 0.2$), the European population had very long LD blocks (>100 kb). The Indian and American populations had intermediate LD decay. Consistently, the Asian population had the highest recombination rate compared to the other three (Fig. 2b). Different LDs and recombination rates found among populations indicate that the frequencies of sexual reproduction varied among populations. In addition, we found that the variations of heterozygosity in *S. polyrhiza* showed a similar pattern with the genomic diversity and recombination rate among four populations (Fig. 2b and Supplementary Fig. 6).

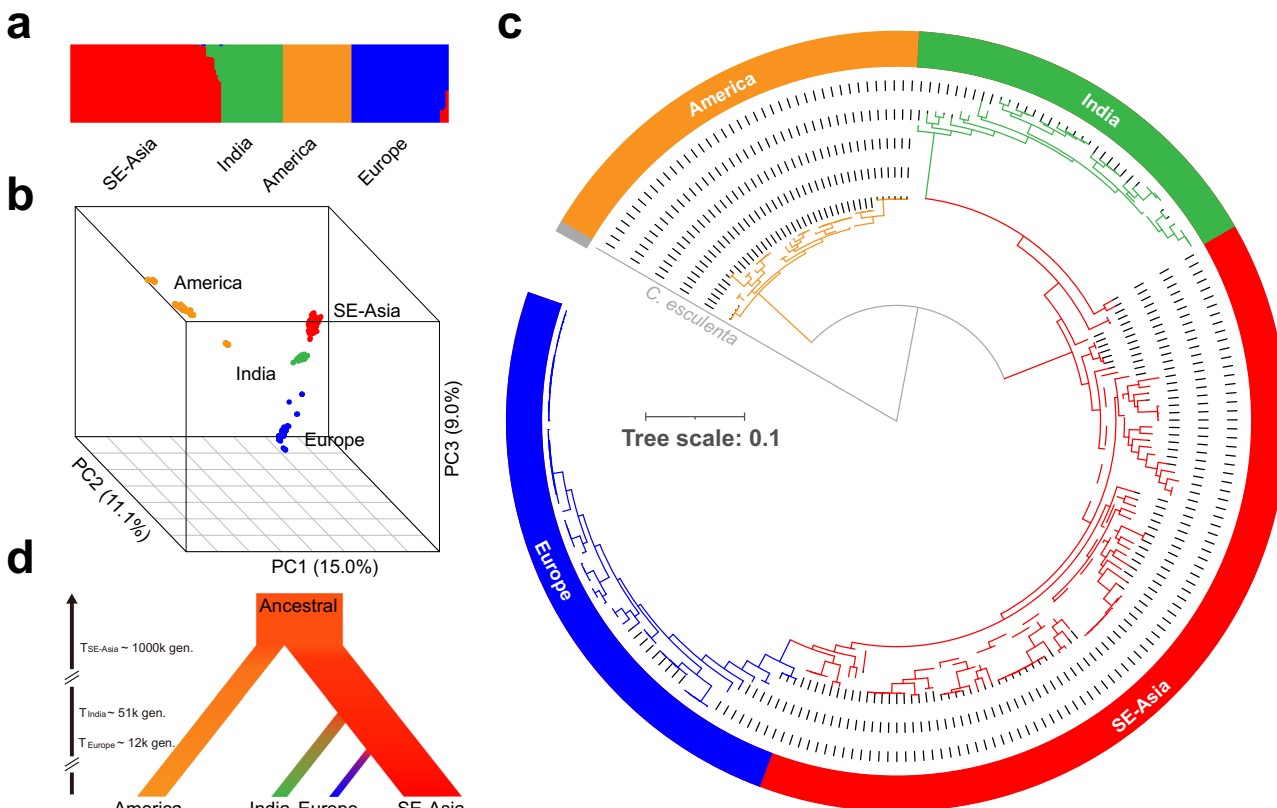

**Fig. 1 | Phylogeny, population structure and demographic model of 228 *S. polyrhiza*. a** The population structure of 228 *S. polyrhiza* genotypes. **b** The principal component (PC) analysis of the SNPs from 228 *S. polyrhiza* genotypes. The three coordinates indicate the first three PCs. **c** The Maximum Likelihood phylogenetic tree of 228 *S. polyrhiza* genotypes. The gray branch represents the outgroup - *C. esculenta*. Dashed branches represent internal nodes with supporting values lower than 0.75 (the max is 1). **d** The demographic model of *S. polyrhiza* populations. "$T_{population}$" indicates the estimated divergence time in generations.

Interestingly, the changes in genomic diversity and levels of heterozygosity are associated with two SVs involving MADS-box genes that are involved in sexual reproduction. One SV is an 84 bp insertion at the last coding sequence (CDS) of gene SpGA2022_005278, a homolog of *AGL62* from the Mα subclade of MADS-box genes (Supplementary Fig. 7). In *A. thaliana*, *AGL62* is a transcription factor that suppresses endosperm cellularization by activating the expression of a putative invertase inhibitor, *InvINH1*, in the micropylar region of the endosperm[52,53] (Supplementary Fig. 8). The insertion may potentially disrupt the function of the *AGL62*-like gene, suggesting a possible reduction in the suppression of endosperm development, which might be required for sexual reproduction (Fig. 2a). Consistently, we found the insertion was at a higher abundance in the SE-Asian population (87.5%) than in other populations (Fig. 2c, d). In addition, the insertion positively correlates with heterozygosity within the European population (Supplementary Table 6 and Supplementary Fig. 9).

Another SV is a 69 bp deletion at 1.8 kb upstream of SpGA2022_007306, (Supplementary Fig. 7), a gene that show homology to *SOC1* (but shorter than *SOC1*, Supplementary Data 7), which is a positive regulator of the flowering process in *A. thaliana*[54] (Fig. 2a). Conserved protein domain analyses suggested that SpGA2022_007306 has *SRF*-like MADS domain but lacks the K-box region (Supplementary Fig. 10), which is similar to *Os03g03100* (*OsMADS50*), a *SOC1* homology that are involved in regulating flowering time in rice[55–58] (Fig. 2a). The deletion was exclusively found in the Indian population with the alternate allele frequency of 73% (Fig. 2c). It is plausible that the deletion, due to its disruption potential at the *cis*-regulatory region, reduces the ability of this *SOC1*-like gene to respond to the upstream floral activators (e.g. *CO*) in *S. polyrhiza*, thus reducing the frequency of sexual reproduction in the Indian population (Fig. 2d). Consistently, this deletion negatively correlates with heterozygosity in the Indian population (Supplementary Table 6, Supplementary Fig. 11). However,

future functional validations on SV of the two MADS-box genes are needed to provide further mechanistic insights into the observed patterns.

## Population epigenomic diversity in *S. polyrhiza*

As changes in sexual reproduction can also alter epigenomic dynamics, we further investigated the patterns of population epigenomic diversity in *S. polyrhiza*. We selected five individuals from each population and quantified their shoot DNA methylation levels at single-base resolution using whole genome bisulfite sequencing (Supplementary Table 7). Similar to a recent study[39], we found that only 1.6% of cytosines are methylated in *S. polyrhiza* (7.6% of CpG, 2.3% of CHG, and 0.1% of CHH; Supplementary Table 8), and the average species-wide methylation level is the lowest among all studied angiosperms (Supplementary Fig. 12)[30,59]. The hierarchical clustering of 20 methylomes in CHG and CHH contexts in gene bodies show overall consistency with their genetic similarity (Supplementary Fig. 13 and 14) with few discrepancies were mostly found within the same population or between the recently diverged SE-Asian and European populations. While in the CpG context, we did not observe clear correlations between genetic and methylation distances (Supplementary Fig. 15).

We then compared the genome-wide weighted methylation level (wML) among populations. For CpG methylation, no differences were found among four populations at genome-wide, gene body, or TE levels (Fig. 3a, d and g). For CHG, the Indian population had the lowest genome-wide methylation level among all four populations (Fig. 3b, e, and h). Interestingly, for CHH, the SE-Asia and Europe populations had the higher genome-wide methylation levels compared to American and India populations (*P* < 0.05, pairwise Wilcoxon test; Fig. 3c), while the European and Indian populations showed a gradual reduction of methylation in comparison to the SE-Asian population. The pattern was the same for both gene

**Fig. 2 | Genomic diversity variation among four populations might result from the switching between sexual and asexual propagation in *S. polyrhiza*. a** Scheme of asexual and sexual propagation cycles in *S. polyrhiza*. (**A**) Vegetative stage of *S. polyrhiza*; (**B**) Budding; (**C**) Offspring from clonal propagation; (**D**) *S. polyrhiza* flowering; (**E**) Putative schematic of ovule at endosperm cellularization stage; (**F**) Putative schematic of seeds. **b** Bar plots show the differences among the four populations in terms of "π": genome-wide nucleotide diversity; "LD": the physical extent (in kb) of pairwise SNPs at $r^2$ of 0.2 (Europe does not yet reach $r^2$ = 0.2 at 100 kb, Supplementary Fig. 5); "$\pi_N/\pi_S$": the efficacy of linked selection; "*r*": genome-wide recombination rate; "H": the median of per population genome-wide heterozygosity rate. **c** Two panels of pie charts indicate the allele frequencies of the functional allele of *SOC1*-like and *AGL62*-like genes among populations. Gray: functional allele frequency; Black: SVs allele frequency. **d** The distribution and migration world map of the four *S. polyrhiza* populations. "+" suggests the increased functional allele frequency, while "-" suggests the decreased functional allele frequency.

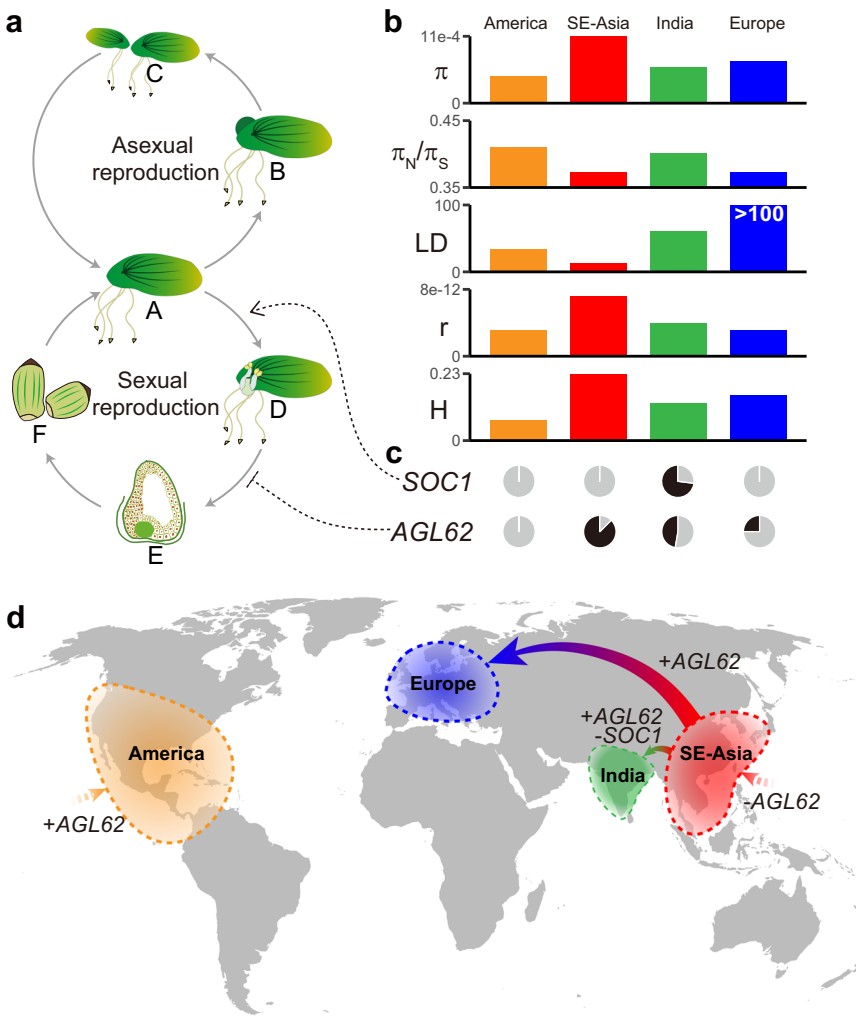

bodies and TEs ($P < 0.05$, pairwise Wilcoxon test; Fig. 3f, i). The genome-wide reduction of CHH methylation is consistent with the hypothesis that clonal reproduction reduces CHH methylation, and the effects gradually accumulate over clonal generations[60].

## The footprint of selection on the genome

To identify the genomic signature of selection at the species level, we performed genome-wide scans. To reduce false positives, we used the μ-statistics from RAiSD[61], the composite likelihood ratio CLR statistic from SweeD[62], and the T statistic from LASSI[63]. We found 69 genes showed strong signatures of selection using all three methods (Supplementary Fig. 16 and Supplementary Data 8). Manual inspection indicated that several orthologs of these genes are related to gametogenesis (e.g., *NOTCHLESS*) and embryogenesis (e.g., *NUP214*, *CPSF*, *CDK*, *AGP*, and *ACR4*) in *Arabidopsis thaliana*[64–68]. Further enrichment analysis indeed showed that embryo lethal genes were enriched in these 69 genes ($P = 0.016$, $\chi^2$ test). In addition, the *A. thaliana* orthologs of several genes under selection are also associated with controlling sexual reproduction, including floral development (*DRMY1* and *ACR4*)[64,69], flowering time (*NF-Y AT2G27470*, *NF-YAT1G72830*, and *CPSF*), pollen development (*EFOP3*, *ELMOD*, and *CLC*)[66,70–74], seed development (*NUP214*, *NF-Y AT2G27470* and *NF-YAT1G72830*, and *Transducin/WD40*)[65,70,75]. Furthermore, among these 69 genes, we also found several genes involved in leaf development and vascularity (*SECA2*, *RbgA*, *PHABULOSA/PHAVOLUTA*)[76–78], light signaling (*NF-Y*, *CCR4-NOT*, and *PPP*)[70,79,80], root development (*GEND1*, *WAVY*, and *ACR4*)[64,81,82], DNA damage repair (*ATM* and *Xrcc3*)[83,84], and stress tolerance (*phospholipase D*, *histone superfamily protein*, *RabGAP*, *FC1*, *NUDX2*) (Supplementary Data 8).

To further understand the selection that drove the evolution within individual populations, we identified the signature of positive selection in a three-population tree using patterns of linked allele frequency differentiation and calculating the corresponding composite-likelihood ratio (CLR, see Methods). In total, we found 1,883 genes on the SE-Asian branch, 593 genes on the Indian branch and 401 genes on the European branch (Fig. 4a; see Supplementary Results Section 2.6, and Supplementary Data 9) which showed strong signatures of selection (top 1% of CLR values). We did not find evidence supporting the hypothesis that differentially methylated genes were under positive selection (see Supplementary Methods Section 1.7, Supplementary Results Section 2.5, and Supplementary Data 10).

We found that genes under positive selection in the European branch are enriched with reproduction and development-related GO terms (Supplementary Fig. 17). Among these, SpGA2022_013448, in chromosome 9, is an ortholog of FLOWERING LOCUS KH DOMAIN (*FLK*) that delays flowering by up-regulating *FLC* family members in *A. thaliana*[85]. This gene showed a strong signature of selection in the European branch but not in other branches (Fig. 4c, d). Similarly, SpGA2022_006111, on chromosome 3, is an ortholog of the *A. thaliana* BIG BROTHER (*BB*) that negatively regulates floral organ size and is also under selection in Europe[86] (Fig. 4c).

In the SE-Asian population, we found that gene SpGA2022_051517, a CHROMOMETHYLASE3 (*CMT3*) ortholog in *A. thaliana* that is likely associated with maintaining CHG methylation[17], was under positive selection. This is consistent with the higher CHG methylation levels observed in the SE-Asian population when compared to the European and Indian populations (Figs. 3a, b). Within the Indian population, we found that five MADS-box genes have been under selection exclusively along this branch. Given that there are 43 MADS-box genes in the genome, the fact that five of

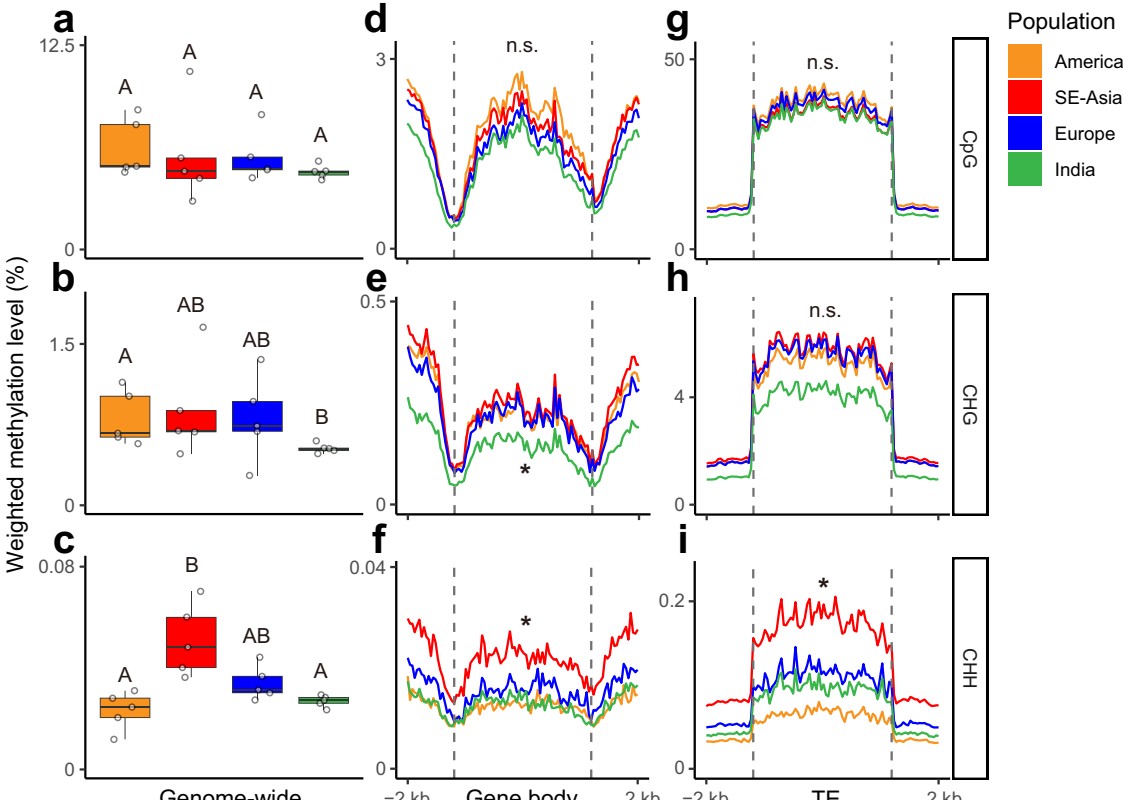

**Fig. 3 | Weighted methylation level (wML) among four populations. a–c** Box plots of genome-wide weighted methylation level (wML) in (**a**) CpG, (**b**) CHG, and (**c**) CHH context (N = 5). Uppercase letters indicate statistical differences among populations using Wilcoxon test (with bonferroni method for multiple tests corrections). **d–f** wML in the gene body and its flanking 2 kb regions in (**d**) CpG, (**e**) CHG, and (**f**) CHH context. **g–i** wML in TE and its flanking 2 kb regions in (**g**) CpG, (**h**) CHG, and (**i**) CHH context. The asterisks indicate significant differences between populations (P < 0.05; Wilcoxon test), while "n.s." indicates no significant difference.

them have been targeted by selection, constitutes a significant enrichment of such genes under selection (P = 0.0075, Fisher's Exact test). For example, SpGA2022_013078 is an homolog of AGAMOUS-LIKE6 (*AGL6*), which is involved in flower and meristem identity specification in rice[87]; SpGA2022_052274, a homolog of *APETALA3* (*AP3*), is involved in the petal and stamen specification in *A. thaliana*[88]; and SpGA2022_006905 belongs to the SHORT VEGETATIVE PHASE (*SVP*-group) which controls the time of flowering and meristem identity[89].

We found 77 genes under positive selection (top 1% CLR values) in both the European and Indian populations (Fig. 4b), significantly more genes than expected by chance (P < 2.2e-16, Fisher's Exact Test). Among these, gene SpGA2022_055195, an ortholog to *CYP78A9* of cytochrome P450 monooxygenases in *A. thaliana*, belongs to a highly conserved gene family CYP78A. Previous studies in *A. thaliana* and other species found that *CYP78A9* plays a critical role in promoting cell proliferation during flower development and further impacts seed size[90–92]. In addition, the RNA-seq data indicates that *CYP78A9* is differentially expressed between India and Europe populations (see Supplementary Methods Section 1.8, Supplementary Results Section 2.7, and Supplementary Data 11). Overall, these data consistently suggest that genes involved in reproduction and development were under selection in Indian and European populations, which might have led to reduced sexual reproduction in these two populations.

## Discussion

Here, we characterized the genomic and epigenomic diversity, as well as the demographic history of a facultative asexual flowering plant, *S. polyrhiza*. We found that among populations of *S. polyrhiza*, demographic history and reproductive system jointly determine the population's genomic and epigenomic diversity. Analyses on the footprint of selection

suggest that natural selection drove the reduced vascular system and increased asexuality in *S. polyrhiza*.

Theory predicts that asexual reproduction reduces genomic diversity and the efficiency of purifying selection[93]. Consistent with this prediction, at the species level, we found that *S. polyrhiza* has very low genomic diversity and reduced purifying selection (seen as an increased $\pi_N/\pi_S$ ratio), when compared to a wide range of spermatophyte plants[42]. Within species, the SE-Asian population, which has the highest frequency of sexual reproduction based on the estimated recombination rate, has the highest genomic diversity, the lowest $\pi_N/\pi_S$ ratio and the highest heterozygosity (Fig. 2b), supporting the theoretical prediction[2,5,13–15]. The low $\pi_N/\pi_S$ ratio found in the European population, which has the lowest sexual reproduction and genomic diversity, is most likely due to its migration history. The demographic model suggested that the European population derived from the SE-Asian population very recently (Fig. 1d). It is likely that the $\pi_N/\pi_S$ ratio in the European population remained the same as its ancestral population and has not reached an equilibrium level yet.

While there are fewer genome-wide SVs in *S. polyrhiza* compared to other species[94,95], we found these variants and small INDELs are in tendency enriched in stress responses and reproduction, such as MADS-box genes. This indicates that the loss-of-function of genes involved in flower development and sexual reproduction, is under natural selection. The results are consistent with the observation that the number of functional MADS-box genes was dramatically reduced in *S. polyrhiza*[49].

Single-base resolution methylomes of 20 individuals showed that the overall CpG, CHG and CHH methylation levels in *S. polyrhiza* shoots are very low, consistent with previous studies[39,41]. The low levels of DNA methylation might be associated with reduced sexual reproduction: while CpG and CHG methylations in plants are important for controlling cross-overs during

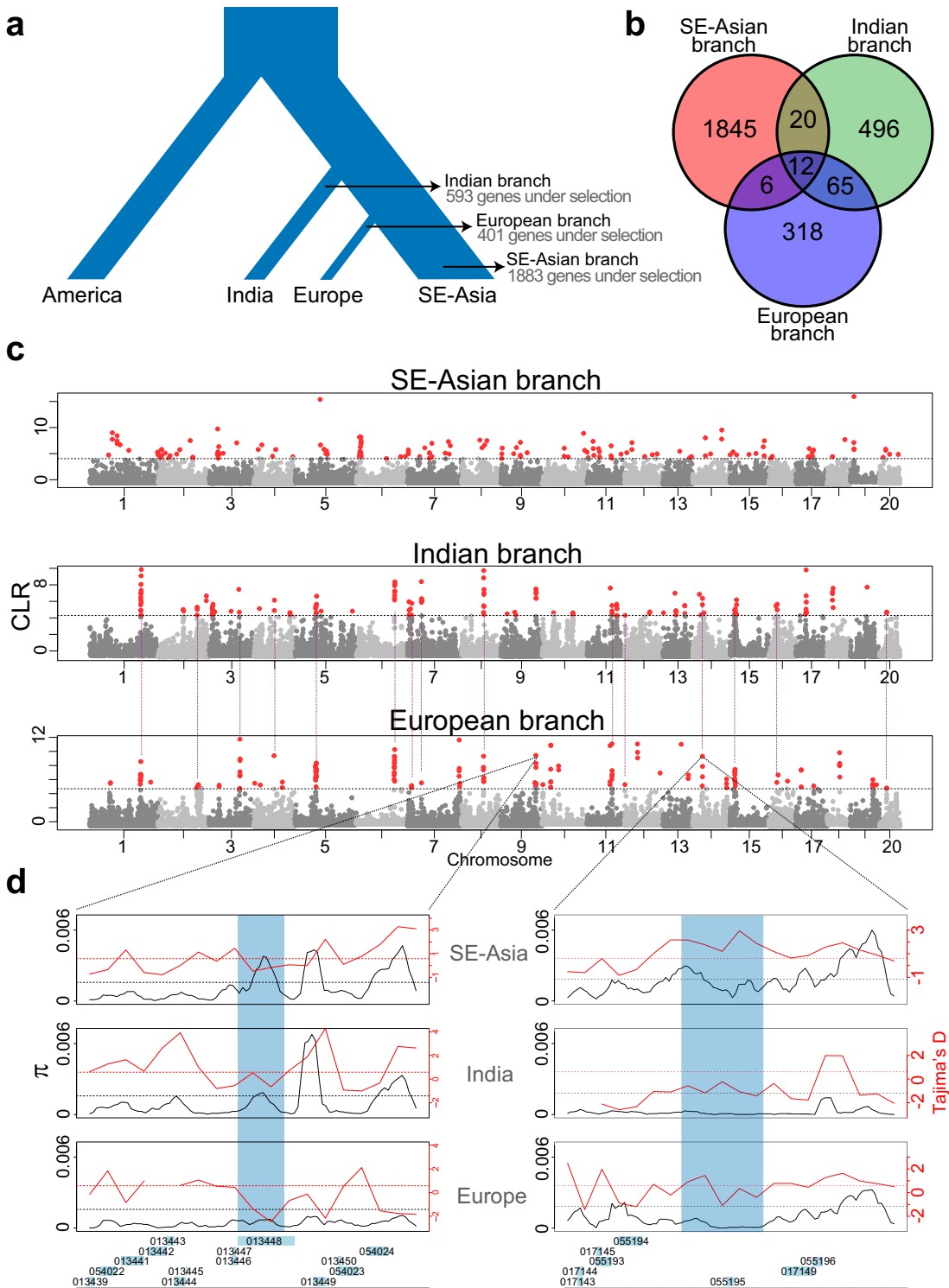

**Fig. 4 | Branch-specific selection signature scans. a** Population tree is used for the population-specific selection analyses. Input consists of an "outgroup" population (America) and two "ingroup" populations (India and SE-Asia, or Europe and SE-Asia). We present the number of genes in the ingroup populations belonging to the top 1% CLR scores reported by 3P-CLR. **b** Venn diagram showing the genes under selection that are common between the populations of SE-Asia, India, and Europe. **c** Branch-specific genome-wide scan of selection: each dot represents a chromosomal locus picked by 3P-CLR for which a CLR score is calculated; loci on the top 1% CLR are shown in red. The dashed purple lines between European and Indian panels indicate common peaks of selection detected at the same genomic regions in both populations. **d** Genomic diversity as depicted by π (solid black lines) and Tajima's D (solid red lines) for two selected loci: the *FLK* (left) and the *CYP78A9* (right). *FLK* shows a signature of selection (low π and negative Tajima's D) in the European population, while *CYP78A9* shows a signature of selection in both Europe and India. Dashed lines indicate genome-wide averages for π (black) and Tajima's D (red).

meiosis[96] and are increased during male gametogenesis, CHH methylation is highly accumulated during embryogenesis[18,24–26]. In facultative asexual plants, due to reduced sexual reproduction and meiosis, the selection of genetic mechanisms maintaining or increasing the CpG, CHG and CHH methylation is reduced or absent, which might have led to the reduced CpG, CHG and CHH methylation levels. Consistently, a recent study suggests that *S. polyrhiza* has lost several genes in the RdDM pathway[41]. Interestingly, within species, the CHG and CHH methylation profile of the 20 individuals largely correlates with their genetic distance (Supplementary Fig. 13 and 14), indicating a gradual neutral evolution of DNA methylomes in *S. polyrhiza*. For example, the Indian and European populations, which diverged from SE-Asian populations around 51,000 and 12,000 generations ago, gradually decreased their CHH methylations (Fig. 3a–c).

At the species level, using a genome-wide scan approach, we found a strong signature of natural selection on genes involved in flower and seed development, indicating that the evolution of reproduction, likely, an increased clonal propagation in *S. polyrhiza*, was driven by natural selection. This is consistent with the pattern that many aquatic organisms reproduce clonally[97]. In addition, several genes related to vascularity, root development and DNA damage repair were also under strong selection, suggesting the reduced root and vascular development and low mutation rate in *S. polyrhiza* were likely also driven by natural selection.

Among populations, we found strong positive selection on genes involved in sexual reproduction and development in India and Europe populations, two recently evolved populations that showed reduced genomic recombination. These results are consistent with the hypothesis that natural selection favors clonal reproduction in *S. polyrhiza* during the recent colonization process, a pattern that was frequently found in many invasive species[98,99]. However, despite strong selection favoring clonal reproduction, substantial recombination in the *S. polyrhiza* genome, mostly in the SE-Asian population, remained, reflecting that sexual reproduction is essential to overcome the costs involved in clonal reproduction in the long term.

Taken together, the structure of population genomes and epigenomes of *S. polyrhiza* suggest that demography and natural selection acting on the reproduction system and organ development can shape genome-wide genomic and epigenomic variations.

## Materials and Methods
### DNA sample preparation and sequencing
We sequenced 131 genotypes that were primarily collected from Asia and Europe (Supplementary Data 1). These samples were cultivated in N-medium[100] until DNA isolation using a CTAB method. Library preparations were carried out following the protocol described in Xu et al.[36]. All libraries were sequenced either on Illumina HiSeq X Ten or Illumina Hiseq 4000 platforms for paired-end sequencing with a read size of 150 bp. Low-quality reads and adapter sequences were trimmed with AdapterRemoval (v2.033)[101]. On average, 33.8 million reads per genotype were obtained. The clean reads were aligned to the *S. polyrhiza* reference genome[48,102] using BWA-MEM (https://github.com/lh3/bwa) with default parameters. Reads without alignment hits or with multiple alignment positions were removed. SAMtools "rmdup" function was used to remove PCR duplicates[103].

### Genetic variant identification and gene family annotation
After filtered out low-quality SNPs using GATK[104] (v4.1.4.1, Java 11) with options: "QD < 2.0 | QUAL < 30.0 | SOR > 3.0 | FS > 60.0 | MQ < 40.0 | MQRankSum < -12.5 | ReadPosRankSum < −8.0", we identified 8,363,387 SNPs. Then, VCFtools (v0.1.13)[105] and GATK were used to remove SNPs that have the following features: (1) SNPs from organelle genomes (9,278 SNPs); (2) missing genotypes >20% (85,645 SNPs); (3) mean sequencing depth <8 or >41 (179,920 SNPs); (4) non-biallelic (448,404 SNPs); (5) minor allele frequency (MAF) <1% (6,102,027 SNPs); and finally, (6) located in small SNP clusters (≥ 3 SNPs in a ten base-pair window, accounted for 296,132 SNPs). We updated the protein-coding gene annotation of *S. polyrhiza* based on recently published transcriptomes and Iso-seq data (see Supplementary Methods Section 1.1, Supplementary Results Section 2.1,

Supplementary Table 9, and Supplementary Figs. 18–20). We used SnpEff (version 5.0c)[106] to annotate SNPs and INDELs. To exam whether SNP cluster filtering criterion affects the estimation of genomic diversity and selection, we performed additional analyses based on a more relaxed filtering parameter (≥200 SNPs in 1 Kb region). Although the second SNP cluster filtering criterion resulted in 18.7% more SNPs, which are mostly (>88%) located in TE regions or nearby the SV or INDELs, the patterns of genomic diversity and selection did not change. In addition to SNPs and INDELs, We identified SVs using a joint genotyping pipeline and stringent quality filtration processes (see Supplementary Methods Section 1.2, Supplementary Results Section 2.2, and Supplementary Fig. 21-24).

We estimated genome-wide nucleotide diversity ($\pi$) and genome-wide $\pi_N/\pi_S$ ratios using SNPgenie (v2019.10.31)[107]. The SNPs overlapping with the structure variations were excluded from the calculation to minimize potential interference caused by misalignments, ensuring a more accurate and reliable analysis.

The genome-wide heterozygosity for each individual was calculated using VCFtools (v0.1.13)[105]. We estimated the genetic associations between heterozygosity and the SVs of *AGL62* and *SOC1* using RVTESTS[108] with the single variant Wald test.

To study the potential genetic factors related to the variation of sexual reproduction frequency in *S. polyrhiza*, we annotated the MADS-box gene family (see Supplementary Methods Section 1.3 and Supplementary Results Section 2.3, Supplementary Fig. 25-27, Supplementary Table 10, and Supplementary Data 12). Other gene families that were annotated in Arabidopsis were also identified in *S. polyrhiza* using an orthology-based method (see Supplementary Methods Section 1.4 and Supplementary Data 4).

### Population structure and linkage disequilibrium (LD)
We grouped genetically similar genotypes by defining clonal genotype pairs that have no more than 0.01% different homozygous sites and no more than 2% different heterozygous sites. These thresholds were previously adopted by Ho et al.[37].

Prior to the population structure analysis, we removed SNPs that (1) deviated from Hardy-Weinberg Equilibrium (Fisher exact test, $P < 0.01$) or (2) linked loci (each pair of SNP have correlation coefficient $r^2 > 0.33$ in a sliding window with a size of 50 SNPs and step of 5 SNPs), using VCFtools (v0.1.13)[105] and Plink (v1.9)[109].

Principal component analysis (PCA) and population structure analysis were carried out using Plink (v1.9)[109] and fastStructure (v1.0)[110], respectively. The simple mode (as default) from fastStructure was used for the population structure analysis. The K value was estimated using a heuristic function in fastStructure.

For each of the 159 clonal families, we selected the least missingness genotype (i.e. the genotype with the highest sequencing coverage of that clonal family) as the representative genotype. SNP information from all 159 representative genotypes was used to estimate the linkage disequilibrium decay for each of the four populations. PopLDdecay (v3.41)[111] was used to measure LD decay. For each population, we used the following filters: SNP of missing allele > 20% and MAF < 0.05. The allele frequency correlation (denoted as $r^2$) of pairwise SNPs within 100 kb physical distance was calculated.

### Phylogenetic tree reconstruction
We used BLAST+ version 2.9.0[112] to identify orthologous fragments between the genomes of *S. polyrhiza* and *Colocasia esculenta* (Araceae). For each SNP from the core set, the reference allele and its flanking 300 bp (upstream 150 bp and downstream 150 bp, respectively) sequences were extracted from the *S. polyrhiza* genome and then aligned to the *C. esculenta* reference genome[113]. The hit thresholds were set as (1) alignment identity >70%; (2) e-value > 1e − 6; (3) minimum aligned sequence length ≥50 (the aligned sequence must cover SNP position); (4) keep the best hit; and (5) ignore short deletions from *C. esculenta*. The orthologous alleles from *C. esculenta* were used as the outgroup genotype. We identified only 13,120 SNPs that have orthologous fragments in the *C. esculenta* genome. Those data were further used to infer the maximum-likelihood

(ML) phylogenetic tree using RAxML-ng (v1.0.1)[114]. The best hit model was estimated to be 'TVM + G4' using Modeltest-ng (0.1.6)[115,116]. The bootstrapping converged after 700 iterations of the ML tree search. ITOL v5[117] and the Python package ETE2[118] were used for tree visualization.

## Selection analysis

Genome-wide scans of selection were performed on all 20 chromosomes of all sampled populations. Selective sweeps were inferred by three programs: RAiSD[61], SweeD[62] and LASSI[63]. RAiSD uses the μ statistic, which provides information on the SFS, LD, and genomic diversity to evaluate the presence of positive selection[61]. SweeD calculates the traditional composite likelihood ratio (CLR) to infer loci under selection[62]. LASSI employs the $T$ statistic, which uses a likelihood model based on the haplotype frequency spectrum to detect hard and soft sweeps[63]. As recommended by the authors of LASSI, we selected the top 5% $T$ scores as candidates for selection. For RAiSD and SweeD we selected the top 1% scores. After finding the common genes under selection according to all three programs, we reported the genes that have orthologs in *A. thaliana*. The embryo lethal genes from *A. thaliana*[119] were used for the enrichment analysis.

To test for population/branch-specific signals of selection, we ran a composite likelihood ratio (CLR) approach as implemented in 3P-CLR[120]. Briefly, this method uses three-population trees coupled with genomic data as input, from which patterns of linked allele frequency differentiation are calculated. By doing this, this algorithm can tell apart signals of selection that happened in either branch of the tree or in the ancestral lineage, as well as outputting the loci with the highest CLR[120]. In our case, we used either a North America-Asia-Europe, or a North America-Asia-India population tree as input, and 3P-CLR output the CLR across windows along each chromosome in the *S. polyrhiza* genome. We then selected the top 1% windows for each branch of the input tree and reported the genes that are present in each window. To further validate the evidence of positive selection on the regions with the highest CLR, we ran scans of Tajima's D and genomic diversity along the same windows and contrasted them with the same signal along the other population branches. We expect negative Tajima's D and low genomic diversity values along the populations with high CLR values. For authenticity validation of genes under selection, we used RT-qPCR to check the expression of eight genes (see Supplementary Methods 1.9, Supplementary Results 2.8, Supplementary Fig. 28, and Supplementary Table 11 and 12). Another expanded list that includes 37 candidate genes was also created, and these genes' expression (RNA-seq) and orthology alignments against their Arabidopsis counterparts were examined (Supplementary Data 7).

## DNA methylation in *S. polyrhiza*

We selected five genotypes from each of the four populations (America, India, SE-Asia, Europe) for single-base whole-genome bisulfite sequencing (WGBS). The genotypes originated from distinct clonal families, except for two European genotypes that came from the same clonal family (Supplementary Table 7).

FastQC (v0.11.5, https://www.bioinformatics.babraham.ac.uk/projects/fastqc/) was used to summarize statistics of the sequencing reads. Trimmomatic (v 0.36)[121] was used to filter out low-quality reads with the parameters "SLIDINGWINDOW: 4:15, LEADING:3, TRAILING:3, ILLUMINACLIP: adapter.fa: 2: 30: 10, MINLEN:36". To account for the genetic variations among genotypes, we generated pseudo-reference genome for each genotype by substituting SNP from the *S. polyrhiza* reference genome using GATK, using a similar strategy to previous studies[122,123]. Bismark (v 0.16.3)[124] was used to align bisulfite-treated reads to pseudo-reference genomes. Identical reads aligned to the same genomic regions were deemed as duplicated reads and thus were removed. Cytosines covered by less than five sequencing reads were excluded from the study. Only after applying these filters the sequencing depth and coverage were then summarized. The sodium bisulfite non-conversion rate was calculated as the percentage of non-converted cytosines to all cytosines in the reads that mapped to the chloroplast genome[125] (GenBank: JN160603.2). For each cytosine site, a binomial test was performed to determine if the cytosine was methylated. If the methylation frequency at the site was lower than the background, which was estimated as the non-conversion rate, then the site was considered unmethylated, and the reads supporting methylation at this site were excluded[126].

We calculated two different methylation parameters: the proportion of methylated cytosines (mC methylation) and weighted methylation level (wML)[126]. For both parameters, only cytosines covered by more than four sequencing reads were involved in the calculation. Those cytosines with low reads supporting methylation but not passing the binomial test were considered as un-methylated cytosines. The mC proportion was calculated by dividing the number of methylated cytosines by the total number of cytosines. Genomic regional wML was calculated using the methylKit (v1.17.5)[127] and the regioneR (v1.28.0)[128], with input based on the cytosine report generated with the Bismark pipeline. Line plots that show the wML patterns across the gene body and transposable elements, as well as their 2 kb flanking regions, were generated using ViewBS (v0.1.11)[129]. The hierarchical clustering, based on the methylation profiles' similarity, was done using methylKit. The comparison between the genetic phylogenetic tree and hierarchical clustering based on the methylome was made using the R packages ggtree (v3.4.4)[130], treeio (v1.20.2)[131], ape (v5.6.2)[132], and phytools (v1.2.0)[133].

## Reporting summary

Further information on research design is available in the Nature Portfolio Reporting Summary linked to this article.

## Data availability

The raw genomic and bisulfite sequencing reads involved in this study can be retrieved from NCBI under accession numbers Bioproject PRJNA701543 and Bioproject PRJNA934173. The scripts for the data analyses are deposited in https://github.com/Xu-lab-Evolution/Great_duckweed_popg. The authors declare that the data and corresponding computational codes supporting the conclusions of this study are available within the article and its supplementary information file.

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

## Acknowledgements

We thank Martin Schäfer, Marie Sárazová, and Laura Böttner for supporting plant sample maintenance and DNA isolations. We thank Arturo Mari-Ordonez and Pavlos Pavlidis for valuable comments and Alex Widmer for contributing resources at the early stage of this project. This project is supported by the German Research Foundation (427577435 and 438887884 to S. X, and 422213951 to M. Hu), the Center for Adaptation to a Changing Environment (ACE) at ETH Zurich (to S. X.), the Swiss National Science Foundation (P400PB_186770 to M. Hu.), the Volkswagen Foundation (97236 to M. Hu.) and through career development measures of the University of Münster (to M. Hu.) The project was inspired by discussions with the members of the CRC TRR 212 (NC3) – Project number 316099922, and Research Training Group 2526 (GenEvo) – Project number 407023052. Parts of this research were conducted using the supercomputer Mogon and/or advisory services offered by the Johannes Gutenberg University Mainz (hpc.uni-mainz.de), which is a member of the AHRP (Alliance for High-Performance Computing in Rhineland Palatinate, www.ahrp.info) and the Gauss Alliance e.V. The authors gratefully acknowledge the computing time granted on the supercomputer Mogon at the Johannes Gutenberg University Mainz (hpc.uni-mainz.de) and PALMA-II at the University of Münster.

## Author contributions

Y. W., P. D. and S. X. performed data analysis. A. C. and M. Ho. performed the experiments. K. J. A., H. Z., K. S. S., and S.X. contributed to the giant duckweed collections and resources. S. X. and M. Hu. conceived and supervised the project. S. X., Y. W., P. D., A. C. and M. Hu. wrote the manuscript. All authors contributed to the final version of the manuscript.

## Funding

## Competing interests

The authors declare no competing interests.
