## [Peer Review File · Communications Biology]

Reviewers' comments:

Reviewer #1 (Remarks to the Author):

The authors selected the *Spirodela polyrhiza* as study model, and conducted population genetics on 131 materials to elucidate their genetic diversity, possible evolutionary pathways, adaptive evolution, as well as genetic variations of key genes (such as AGL62). Combined with DNA epigenetic analysis on representative individuals, the authors tried to elucidate the genetic mechanism of asexual reproduction in this species. The main results are as follows: there are four populations, America, South East Asia (SE Asia), Europe, and India, with different genetic characteristics. Genome-wide scans revealed that multiple genes involved in flowering and embryogenesis were under positive selection, consistent with the hypothesis that natural selection drove the evolution of asexuality during the recent habitat expansions in this plant. Within species, demographic history and the frequency of asexual reproduction jointly determined intra-specific variations of genomic diversity and DNA methylation levels. There is still a question that authors need to clarify. This article uses *Arabidopsis* as a reference and mentions homologous genes and orthologous genes. How do you define them? Please provide an explanation in the method. In addition, this article identifies candidate genes involved in asexual reproduction at the DNA level through population analysis and homology comparison. Can you select some for quantitative analysis by qRT-PCR to detect their authenticity? This study contributes to the field of asexual reproduction. The manuscript is also well-written. Therefore, I think the manuscript can be acceptable after revisions.

Reviewer #2 (Remarks to the Author):

Wang et al. present extensive data on genetic diversity and its association with selection on genes involved in reproduction and several aspects of plant development. For reasons outlined in my detailed comments below, I strongly recommend that the authors temper their claims about recombination (and by inference sexual reproduction). While their claims may be correct, there is at least one other explanation for them that may have nothing to do with the frequency of recombination (or sexual reproduction). More importantly, the central claim of this paper is that "multiple genes involved in flowering and embryogenesis were 38 under positive selection, consistent with the hypothesis that natural selection drove the evolution 39 of asexuality during the recent habitat expansions in this plant" (quoted from the abstract). There are two problems with this claim.

1) The authors acknowledge in their concluding paragraph that genes showing evidence of natural selection included those "involved in flower and seed development" While it's true that only Indian and European populations showed evidence of strong positive selection on genes involved in sexual reproduction, the failure to find positive selection on other genes in these or other populations does not

mean that it's absent, especially since species-wide analyses also revealed evidence for positive selection at genes involved in leaf development and vascularity, root development, and DNA damage repair. Thus, the focus on flowering and embryogenesis in the abstract is inappropriate, and the statement in the concluding paragraph, while broader than the statement in the abstract, does not faithfully reflect the full range. of genes in which positive selection was detected.

2) The authors fail to acknowledge that the patterns of diversity they find could have explanations other than differences in the frequency of recombination (which they take as a marker of the frequency of sexual reproduction. This is both because the patterns themselves could be artefactual (ll. 162-173) and because the **associations** they find may not be causal (ll. 325-327).

Because of these deficiencies, I cannot recommend publication of this paper in its current form. That being said, the deficiencies I describe are correctable, and once corrected, a revised paper could be a valuable addition to the literature. I leave it to the editors to determine whether a revised version should be considered for publication.

ll. 58-59: Strictly speaking "changes between sexual and asexual reproduction" DON'T profoundly affect fitness. Or at least changes don't affect individual fitness, i.e, the probability of survival and reproduction and the number of offspring produced. What it affects is the ability to persist in the short term and in the long term.

l. 74: If the authors are going to refer to a megaspore mother cell, which is reasonable, why not also refer to a microspore mother cell (instead of "its male analog").

l. 111: How were neutral sites identified? Were they SNPs at 3rd positions (in coding sequences) and SNPs in non-coding sequences. The authors should say briefly how they were identified or if they are assuming that all of the SNPs they identified are neutral. And to be precise they should write that the SNPs are **presumed** to be neutral. By the way, while the dN/dS ratio might be relatively high, it is still substantially less than 1 meaning that many of the SNPs must be under selection.

ll. 112-120: Do "multicellular eukaryotes" and "all reported multicellular organisms" include bdelloid rotifers? I know that they are thought to be almost entirely asexual, and I believe that they show extremely low levels of nucleotide diversity.

ll. 120-122: I must be missing it, but I don't see anything about overlap in MADS-box genes in the supplementary information.

ll. 131-135: It's worth noting that population structure and principal component analyses reveal structure **in the data that were collected**. The structure that is revealed depends heavily on sample size. Had the authors analyzed 1300 genomes (instead of 131), for example, there's a very good chance

that they would have identified. more than four clusters. That the authors identified four clusters in this study and in their previous study is less meaningful than the identified clusters (a) reflect geographical grouping of the individual samples and (b) correspond between the two analyses. These results give us confidence that at the level of genetic discrimination available given the sample size, the geographical structure revealed in the data is characteristic of the populations from which samples were collected.

II. 136-145: I don't see any details on how ABC was used to construct the population phylogeny. This needs to be explained.

II. 146-154: The authors should recognized that their inference of population history is based on choosing from one of only three possible scenarios. The scenario they present is the best supported, but there could (in principle at least) be a scenario they did not consider that would be even better supported. Only if they have good reason to believe that the three scenarios they consider are the only ones that are biologically plausible are the conclusions here reasonable.

II. 162-173: LD can also arise from geogrqaphic structure within a sample. The "populations" in which LD was calculated may well consist of many distinct subpopulations (which is related to my comments on II. 131-135). If so, then the apparent rapid decay of LD in Asian populations may simply be an artifact of higher genetic diversity rather than a difference in recombination rates, i.e., rates of sexual reproduction.

II. 174ff: The association of heterozygosity with the MADS-box gene SVs is interesting, but high heterozygosity isn't necessarily a marker of sexual reproduction. Completely asexual populations may be both genotypically uniform and highly heterozygous, since the offspring of asexually produced heterozygotes will never be homozygous.

II. 325-327: The authors present good evidence for selection on genes related to flower and seed development, genes related to vascularity, root development, and DNA damage repair. We do not have good evidence that selection on any or all of those sets of genes are responsible for the observed differences in genetic and epigenomic diversity. There are undoubtedly significant differences in ecology and population dynamics in the region. The differences in ecology and evolution might well be responsible for the differences in genetic and epigenomic diversity. What the authors present is evidence for an ****association****, not a causal relationship.

Reviewer #3 (Remarks to the Author):

The study reveals that natural selection drives the evolution of asexuality in the plant *Spirodela polyrhiza*, leading to low genomic diversity and DNA methylation levels, during habitat expansions.

There are several points that author should address.

1. The authors conclude that *Spirodela polyrhiza* has low genomic diversity, but the stringency of the SNP filtering criteria in method section raises questions about this conclusion. Specifically, the MAF threshold could potentially remove true SNPs, thereby affecting the diversity measurements. The paper should provide detailed information on the number of SNPs removed at each filtering step to substantiate the claims about low genomic diversity.
2. Additionally, the requirement for SNPs to be biallelic is not adequately justified in the paper. Biallelic filters can restrict the scope of genetic diversity by excluding multiallelic variants, which may be informative for understanding the plant's genomic landscape. An explanation for why only biallelic SNPs were considered would strengthen the study's methodology.
3. Ln. 121 : The focus on MADS-box genes for the association with structural variations (SVs) is narrow for a genome-wide analysis. While these genes may be relevant, exclusively concentrating on them risks overlooking the role of other gene families in explaining the asexuality of *Spirodela polyrhiza*. A rationale for why MADS-box genes were specifically selected for this analysis should be provided.
4. LN. 156: Before concluding on the recent species split, the authors should consider confounding factors like population size, migration, and genetic drift that could influence both the nucleotide diversity (π) and the non-synonymous to synonymous substitution ratio (π_N/π_S). These factors can confound the interpretation of genomic data.
5. LN 162: While the paper attributes variations in linkage disequilibrium (LD) and recombination rates among populations to differing frequencies of sexual reproduction, it's important to also consider the impact of mutation rates on these metrics. Mutation rate is another key factor that can influence both LD and recombination rates and should be accounted for in the analysis.
6. LN 174 : The repeated focus on MADS-box genes for their association with changes in genomic diversity and levels of heterozygosity is limiting. A genome-wide or gene-family-wide analysis would offer a more comprehensive view and could potentially identify other significant genes involved in sexual reproduction. Relying solely on candidate gene analysis like MADS-box genes restricts the scope and may miss other relevant factors. This section can be merged with the last section of "Footprint of selection on the genome" and explain why MADS-box is important.
7. The epigenomic analysis presented in the study is somewhat cursory. For a more thorough understanding of gene variation selection history and methylation patterns, it would be beneficial to integrate this analysis with a genome-wide footprint section. This would allow for a more comprehensive examination of how methylation patterns affect gene bodies or upstream regions with selection pressures, thereby providing a richer context for interpreting the data.
8. Lastly, it is imperative that all NGS raw reads be uploaded to a public repository and made accessible to reviewers. This is crucial for the verification and reproducibility of the study's findings.

Reviewer #4 (Remarks to the Author):

Wang et al. conduct a population scale study of genetic and DNA methylation variation in duckweed indicating that the history of asexual reproduction in this species has had genome-wide genetic and epigenomic effects. This is fitting with other studies by this group and others. Genome scans and analysis of genetic variants identifies a number of interesting candidate mutations and genes that affect sexual reproduction and that may underly reproductive differences in these populations. My major issue with

this paper is that the authors put forward a number of very strong claims often on weak evidence. More care should be taken to present results in accordance to the level of evidence at hand.

1. Line 66: “epigenetic markers”, recommend changing to “chromatin marks”
2. Line 66: use “e.g.” instead of “i.e.” as there are other chromatin features besides DNA methylation
3. Line 66 “DNA methylations” should be “DNA methylation”
4. Lines 69-70: As currently written, it makes it sound as if CpG and CHG can be maintained by either MET1 or CMT3 interchangeably. I recommend modifying this sentence to “CpG and CHG methylation are maintained by methyltransferases1 (MET1) and CHROMOMETHYLASE3 (CMT3), respectively,”
5. Lines 70-71: RdDM is capable of methylating all sequence contexts, not just CHH (see Matzke & Mosher 2014 Nature Reviews Genetics).
6. Lines 71-72: The cited paper does not support a role of CMT3 in maintaining CHH methylation. Rather the authors found hypermethylation of CHH at the Onsen transposon due to compensation by CMT2 in *cmt3* mutants.
7. Line 101: “genomic variations” should be “genomic variation”
8. Line 110-112: The authors claim that the genome-wide nucleotide diversity at neutral sites is the lowest amongst multicellular organisms. However, they provide no citations to support this nor do they provide any data from other species that would demonstrate this. The evidence provided by the authors for this claim is insufficient.
9. Line 118-119 & 290-291: The authors argue that the “abundance of the SVs is substantially lower than all reported multicellular organisms.” They only cite two papers for this claim. The first is for grapevine and the second is for peach. Structural variation has been studied in many multicellular organisms and the authors do not provide sufficient evidence for their claim. Grapevine in particular is a not a good choice for comparison as it is a highly heterozygous species that has accumulated extensive deleterious mutations due to its long history of clonal propagation.
10. Line 120-122: The authors claim the MADS box proteins have a higher overlap with SVs. However, the reported p-value is 0.063 which would be considered not-significant by most standards.
11. Line 124-125: The authors argue that the SVs indicate either positive or relaxed selection on the MADS box proteins. I do not find the evidence convincing. First, the first result of overlap with SVs is not significant. Secondly, we are given no details as to the potential effects of these variants. Supplementary Table 2 appears to lump the SVs with the SNPs and includes upstream & downstream regions. Its impossible to ascertain what the actual impact of the SVs are on genes, let alone on the MADS box genes.
12. Line 180-182: The authors state: “The insertion likely disrupted the function of the AGL62-like gene, thus reducing the suppression of endosperm development, which is likely required for sexual reproduction (Fig. 2a).” This is certainly an interesting hypothesis, but it is also speculation and no experimental data is given to show that AGL62 has the same function in duckweed or that this is indeed the effect of the variant. I would highly recommend that the authors tone down the language and instead offer it as a hypothesis.
13. Line 205: “epigenomic variations” should be “epigenomic variation”
14. Line 249 & 251. The authors claim that CMT3 is involved with maintaining CpG methylation. I do not understand their argument. No functional data is given to demonstrate that CMT3 is maintaining CpG. Different chromatin pathways interact and compensate for one another, such that when one particular pathway is affected, there can be downstream effects. A good example of this is the aforementioned

study on the Onsen transposon, where *cmt3* mutations result in compensatory changes in the activity of CMT2. There are also other known indirect relationships between CMT3 and CpG methylation, such as the relationship of CMT3 with CpG gene body methylation. However, there is no direct functional evidence that CMT3 is involved in the maintenance of CpG methylation. The authors should be more cautious in the claims they put forward.

Assignment of reviewers' comments

Reviewers' comments:

Reviewer #1 (Remarks to the Author):

The authors selected the *Spirodela polyrhiza* as study model, and conducted population genetics on 131 materials to elucidate their genetic diversity, possible evolutionary pathways, adaptive evolution, as well as genetic variations of key genes (such as AGL62). Combined with DNA epigenetic analysis on representative individuals, the authors tried to elucidate the genetic mechanism of asexual reproduction in this species. The main results are as follows: there are four populations, America, South East Asia (SE Asia), Europe, and India, with different genetic characteristics. Genome-wide scans revealed that multiple genes involved in flowering and embryogenesis were under positive selection, consistent with the hypothesis that natural selection drove the evolution of asexuality during the recent habitat expansions in this plant. Within species, demographic history and the frequency of asexual reproduction jointly determined intra-specific variations of genomic diversity and DNA methylation levels.

There is still a question that authors need to clarify. This article uses *Arabidopsis* as a reference and mentions homologous genes and orthologous genes. How do you define them? Please provide an explanation in the method.

We appreciate your feedback and the chance to clarify the methods we used to identify 'ortholog' and 'homolog'. In our paper, we mentioned both 'orthologous genes' and 'homologous genes.' To identify orthologous genes between *S. polyrhiza* and *A. thaliana*, we used the one-to-one ortholog identification tool from Hernández-Salmerón et al., 2020. Gene pairs that had a reciprocal best hit between *S. polyrhiza* and *A. thaliana* were defined as orthologs. We have included a detailed description in the additional methods (Supplementary Information section 2.1).

Regarding homologous genes, we focused on the MADS-box gene family in *S. polyrhiza*. We analyzed these genes based on both their sequence similarity and phylogenetic relationships (for more details, please refer to Supplementary Information, additional methods 2.2). For each clade of the MADS-box family, there are multiple homologous copies, making it hard to find the real orthologous gene pairs between two species. We then decided to use the term 'homology' rather than 'orthology' to describe these genes, as it better reflects their relationships.

Reference:

Hernández-Salmerón, Julie E., and Gabriel Moreno-Hagelsieb. "Progress in quickly finding orthologs as reciprocal best hits: comparing blast, last, diamond and MMseqs2." *BMC genomics* 21 (2020): 1-9.

In addition, this article identifies candidate genes involved in asexual reproduction at the DNA level through population analysis and homology comparison. Can you select some for quantitative analysis by qRT-PCR to detect their authenticity?

To validate the candidate genes which are potentially related to asexual reproduction through quantitative analysis, we have performed quantitative real-time PCR (qRT-PCR) on eight selected genes from two different genotypes (Supplementary Information 1.8 and Additional methods section 2.9). Additionally, we also provide RNA-seq data from shoot tissues of 19 representative genotypes selected from four populations (Supplementary Information 1.7, Additional methods section 2.8 and Supplementary Data 7). In our supplementary data, we have provided a detailed table and the corresponding expression plot for the genes analyzed in the qPCR experiment (Supplementary Table 12 and Supplementary Fig. 28).

In the qPCR experiment, six genes showed expression in fronds tissue in both genotypes, confirming their authenticity. Two genes (AGL62 and AGL6) showed no expression (Supplementary Table 13) in the frond tissue, likely due to their specific expression during embryogenesis and thus potentially having specific expression in flower tissues (H. Kang et al. 2008; H. F. Li et al. 2010).

Furthermore, the RNA sequencing data has provided us with a broader perspective, revealing that the expression patterns of these eight genes are highly consistent with the qRT-PCR results (Supplementary Table 13), with one exception of the gene SpGA2022_052159 (ACR4). The discrepancy for ACR4 expression could be attributed to the differences of sensitivity and specificity between qRT-PCR and RNA-seq technologies.

Together, the combined approach of qRT-PCR validation and RNA-seq analysis across multiple genotypes and populations provides robust evidence for the authenticity and relevance of the candidate genes.

Reference:

H. Kang, J. G. Steffen, M. F. Portereiko, A. Lloyd, G. N. Drews, The AGL62 MADS domain protein regulates cellularization during endosperm development in Arabidopsis. *Plant Cell* 20, 635-647 (2008).

H. F. Li et al., The AGL6-like gene OsMADS6 regulates floral organ and meristem identities in rice. *Cell Res.* 20, 299-313 (2010).

This study contributes to the field of asexual reproduction. The manuscript is also well-written. Therefore, I think the manuscript can be acceptable after revisions.

Thanks.

Reviewer #2 (Remarks to the Author):

Wang et al. present extensive data on genetic diversity and its association with selection on genes involved in reproduction and several aspects of plant development. For reasons outlined in my detailed comments below, I strongly recommend that the authors temper their claims about recombination (and by inference sexual reproduction). While their claims may be correct, there is at least

one other explanation for them that may have nothing to do with the frequency of recombination (or sexual reproduction). More importantly, the central claim of this paper is that "multiple genes involved in flowering and embryogenesis were 38 under positive selection, consistent with the hypothesis that natural selection drove the evolution 39 of asexuality during the recent habitat expansions in this plant" (quoted from the abstract). There are two problems with this claim.

1) The authors acknowledge in their concluding paragraph that genes showing evidence of natural selection included those "involved in flower and seed development" While it's true that only Indian and European populations showed evidence of strong positive selection on genes involved in sexual reproduction, the failure to find positive selection on other genes in these or other populations does not mean that it's absent, especially since species-wide analyses also revealed evidence for positive selection at genes involved in leaf development and vascularity, root development, and DNA damage repair. Thus, the focus on flowering and embryogenesis in the abstract is inappropriate, and the statement in the concluding paragraph, while broader than the statement in the abstract, does not faithfully reflect the full range. of genes in which positive selection was detected.

We agree with the reviewer. Indeed, we acknowledge that changes between sexual/clonal reproduction isn't the only process that has influenced the evolutionary/adaptive history of this species. Although we focus on the impact of genome-wide selection on reproductive types, we agree that other processes are also important. We have now revised the abstract and manuscript texts accordingly (lines 36-39, and lines 323-333).

2) The authors fail to acknowledge that the patterns of diversity they find could have explanations other than differences in the frequency of recombination (which they take as a marker of the frequency of sexual reproduction. This is both because the patterns themselves could be artefactual (ll. 162-173) and because the **associations** they find may not be causal (ll. 325-327).

Yes, we agree with the reviewer that recombination isn't the only driver of the diversity we see, but the demographic history of the species also plays an important role. For this reason, in this study, we paid close attention to the demography of *S. polyrhiza* and estimated population sizes, colonization times and migration rates, all of which have a direct influence on the genomic diversity. We have now modified lines 293-298 to account for all this. We also reworded the conclusions and acknowledged the non-causal explanations.

Because of these deficiencies, I cannot recommend publication of this paper in its current form. That being said, the deficiencies I describe are correctable, and once corrected, a revised paper could be a valuable addition to the literature. I leave it to the editors to determine whether a revised version should be considered for publication.

Thank you for your critical feedback. We now have addressed the concerns in the revision, which significantly improved the manuscript.

ll. 58-59: Strictly speaking "changes between sexual and asexual reproduction" DON'T profoundly affect fitness. Or at least changes

don't affect individual fitness, i.e, the probability of survival and reproduction and the number of offspring produced. What it affects is the ability to persist in the short term and in the long term.

Agree. We have now reworded that sentence (lines 57-60).

I. 74: If the authors are going to refer to a megaspore mother cell, which is reasonable, why not also refer to a microspore mother cell (instead of "its male analog").

Thank you for pointing out this issue. We agree that it would be more precise to refer to microspore mother cell directly. We have revised the sentence as 'In both male and female gametogenesis, the megaspore mother cell and microspore mother cell experience dramatic chromatin changes during cell specification, such as heterochromatin decondensation and an enlarged nuclear volume.' (line 71)

I. 111: How were neutral sites identified? Were they SNPs at 3rd positions (in coding sequences) and SNPs in non-coding sequences. The authors should say briefly how they were identified or if they are assuming that all of the SNPs they identified are neutral. And to be precise they should write that the SNPs are ****presumed**** to be neutral. By the way, while the dN/dS ratio might be relatively high, it is still substantially less than 1 meaning that many of the SNPs must be under selection.

Thank you for bringing our attention to this issue. Firstly, regarding the identification of neutral sites in our study, we utilized SNPgenie (see Methods) to calculate nonsynonymous and synonymous genomic diversity. This helped us in identifying the SNPs located in the protein-coding regions. In our study, we have considered synonymous SNPs as neutral SNPs based on the common assumption that synonymous mutations do not alter the amino acid sequence of the protein and are therefore likely to be selectively neutral. We realize that we did not provide a clear definition of neutral sites in our manuscript, which might have caused the confusion. Now, we revised our manuscript accordingly (lines 108-113)

We agree that a π_N/π_S ratio of 0.37 indicating many SNPs are under purifying selection, and it is lower than that of many other species. We noted that the value of π_N/π_S ratio in the original version was reported as 0.375 instead of 0.37. In the revision, we now corrected it, see line 111.

II. 112-120: Do "multicellular eukaryotes" and "all reported multicellular organisms" include bdelloid rotifers? I know that they are thought to be almost entirely asexual, and I believe that they show extremely low levels of nucleotide diversity.

In light of your comments, we now revised the sentence to " Consistent with our previous study(36), the genome-wide nucleotide diversity is 0.0016 (Supplementary Table 1), which falls within the lower range of genome-wide nucleotide diversity of other tested multicellular eukaryotes (Supplementary Table 2 and Supplementary Fig. 2)." (lines 108-113). We have removed the sentence "The abundance of the SVs is substantially lower than all reported multicellular organisms." since we don't have enough data to support this statement and this observation is not relevant to the main topic of our manuscript.

Furthermore, to address your point regarding the nucleotide diversity data of Bdelloid rotifers, it is to our current knowledge that there is no published data on the genomic diversity of the natural population of Bdelloid rotifers.

II. 120-122: I must be missing it, but I don't see anything about overlap in MADS-box genes in the supplementary information.

Sorry for the wrong labelling. Supplementary information section 1.3 only described additional results of structure variation analysis in *S. polyrhiza*. The methods we used to test the enrichment between SVs and INDELS with different gene families were detailed in the Supplementary information section 2.4 and 2.5. For all the SVs annotation information (including SV overlapped with MADS-box), we placed this information in an additional Supplementary Data 3.

II. 131-135: It's worth noting that population structure and principal component analyses reveal structure ****in the data that were collected****. The structure that is revealed depends heavily on sample size. Had the authors analyzed 1300 genomes (instead of 131), for example, there's a very good chance that they would have identified more than four clusters. That the authors identified four clusters in this study and in their previous study is less meaningful than the identified clusters (a) reflect geographical grouping of the individual samples and (b) correspond between the two analyses. These results give us confidence that at the level of genetic discrimination available given the sample size, the geographical structure revealed in the data is characteristic of the populations from which samples were collected.

Thank you for pointing out this issue. We agree that larger samples can potentially reveal more clusters. To ensure our text makes clear our groupings come only from the sample (of 228 genotypes) we have now changed the sentence as follows: "Population structure and principal component analyses revealed four populations in the sampled *S. polyrhiza* (Fig. 1a and 1b)." (lines 133-134)

II. 136-145: I don't see any details on how ABC was used to construct the population phylogeny. This needs to be explained.

Only maximum likelihood was used for the phylogeny. ABC was used to test different models of population histories. We rephrased the sentences along lines 138-140 to make this clear.

II. 146-154: The authors should recognize that their inference of population history is based on choosing from one of only three possible scenarios. The scenario they present is the best supported, but there could (in principle at least) be a scenario they did not consider that would be even better supported. Only if they have good reason to believe that the three scenarios they consider are the only ones that are biologically plausible are the conclusions here reasonable.

Indeed, this is true for all types of Bayesian inference in general. The method estimates the best fit only among the tested models. In our case, we are certain that European and Indian populations are more derived based on phylogenetic analysis, therefore we tested only for the most probable ancestral population. We rephrased the text in the revised manuscript.

II. 162-173: LD can also arise from geographic structure within a sample. The "populations" in which LD was calculated may well consist of many distinct subpopulations (which is related to my comments on II. 131-135). If so, then the apparent rapid decay of LD in Asian populations may simply be an artifact of higher genetic diversity rather than a difference in recombination rates, i.e., rates of sexual reproduction.

The reviewer is right in pointing out that the level of structure in populations directly affects LD. However, in our case all studied populations are highly structured at similar levels, at least according to the fastSTRUCTURE analysis performed (Fig. 1a). We ran this analysis also for up to $K=10$. If there were more subpopulations influencing LD then they would have shown up in that analysis.

We believe the rapid decay of LD in Asian population is less likely simply due to higher genetic diversity, as the genome-wide purifying selection in Asian populations are also stronger than others (as indicated by π_N/π_S), which is consistent with the theoretical predictions. We now added cautions in the discussion.

II. 174ff: The association of heterozygosity with the MADS-box gene

SVs is interesting, but high heterozygosity isn't necessarily a marker of sexual reproduction. Completely asexual populations may be both genotypically uniform and highly heterozygous, since the offspring of asexually produced heterozygotes will never be homozygous.

Thank you for bringing our attention to the complex relationship between heterozygosity and reproductive mode. We fully recognize that the association between high heterozygosity and sexual reproduction is not strictly causal. First, we appreciate your point that asexual populations can be both genotypically uniform and highly heterozygous. However, it's worth noting that asexually reproducing individuals can lose ancestral heterozygosity through mechanisms such as loss-of-heterozygosity (LOH), which is not uncommon during the asexual generation. Consequently, under certain circumstances, a high degree of heterozygosity could be suggestive of a higher rate of sexual reproduction. Secondly, in this study, we observed a consistent manner of the levels of heterozygosity with recombination rate, selection, and LD decay across four populations. The synchronized distribution trend of these genetic parameters provides certain rationality of using heterozygosity as an indirect proxy for estimating sexual reproduction rate, especially in the absence of direct methods for measuring the sexual reproduction rate in *S. polyrhiza*. We completely agree that caution must be taken (as we did in our manuscript) in making robust conclusions about inferring different reproductive modes based on the levels of heterozygosity.

II. 325-327: The authors present good evidence for selection on genes related to flower and seed development, genes related to vascularity, root development, and DNA damage repair. We do not have good evidence that selection on any or all of those sets of genes are responsible for the observed differences in genetic and epigenomic diversity. There are undoubtedly significant differences in ecology and population dynamics in the region. The differences in ecology and evolution might well be responsible for the differences in genetic and epigenomic diversity. What the authors present is evidence for an **association**, not a causal relationship.

We agree with the reviewer that we present evidence of an association. We do not mention any direct causal relationships in this manuscript. In our view, selection, evolution and ecology are not mutually exclusive, but they all interact with one another. In the revision, we toned down several conclusions.

Reviewer #3 (Remarks to the Author):

The study reveals that natural selection drives the evolution of asexuality in the plant *Spirodela polyrhiza*, leading to low genomic diversity and DNA methylation levels, during habitat expansions.

There are several points that author should address.

1. The authors conclude that *Spirodela polyrhiza* has low genomic diversity, but the stringency of the SNP filtering criteria in method section raises questions about this conclusion. Specifically, the MAF threshold could potentially remove true SNPs, thereby affecting the diversity measurements. The paper should provide detailed information on the number of SNPs removed at each filtering step to substantiate the claims about low genomic diversity.

We understand your concerns about the potential removal of true SNPs with our minor allele frequency (MAF) filtering process, which could affect the measurements of genomic diversity in *S. polyrhiza*. To further substantiate our filtering criteria, we have conducted additional analysis comparing the number of SNPs with and without MAF filtering. Specifically, we show a table below with the detailed counts of SNPs from each filtering step.

	MAF_filtration		no_MAF_filtration	
	SNPs to remove	proportion	SNPs to remove	proportion
	Basic filtration for "core set"			
number of clean SNPs	8363387	100%	8363387	100%
organelle genomes	9278	0.11%	9278	0.11%
depth < 8 or > 41	179920	2.15%	179920	2.15%
missing genotypes > 20%	85645	1.02%	85645	1.02%
non-biallelic	448404	5.36%	448404	5.36%
MAF < 0.01	6102027	72.96%		
SNP clusters	296132	3.54%	3722798	44.51%
SNP number in the "core set"	1241981	14.85%	3917342	46.84%
	Further filtration for the genomic diversity calculation			
SNPs overlap with SV	73561	0.88%	126421	1.51%
SNPs on the pseudo chromosome	111555	1.33%	296318	3.54%
SNP number for calculating π	1056865	12.64%	3494603	41.78%
genome-wide π	0.0016		0.0013	

Our analysis demonstrates that after filtering out 73% of low-MAF SNPs, the subsequent step of removing clustering SNPs accounted for an additional 3.54% reduction. In contrast, when MAF filtering was not applied, a significantly higher proportion of clustering SNPs (44.5%) were removed. This indicates that many low-MAF SNPs were presented as SNP clusters, likely due to artefacts.

We believe the removal of this amount of low-MAF SNPs is justified, due to the following reasons: 1) We recognize that low-MAF SNPs could potentially be false positives, which might confound subsequent genomic analyses (Linck, E. and Battey, C.J., 2019). 2) Even if these represent true rare mutations, their informational contribution is limited but of high computational cost. 3) In addition, we found that a large number of low-MAF SNPs overlap with SNP clusters (see table above). Our previous study in *S. polyrhiza* showed that such SNPs were false positives (Xu *et al.*, 2019) based on large number of Sanger-sequencing efforts. This is also supported by other studies (e.g. Pfeifer, S. P., 2017), which advocate for the removal of SNP clusters. Our observation that low-MAF SNPs are heavily involved in these clusters corroborates this view. Therefore, we believe that filtering out low-MAF SNPs together with the removal of SNP clusters is a valid approach. This method effectively removes potential sequencing artifacts, ensuring that our analysis is not skewed by erroneous data.

Moreover, we found that even if some low-MAF SNPs were true variants, their impact on the overall estimation of π value was minimal. Our data indicate that even with the inclusion of low-MAF SNPs, the computed π value only slightly changes from 0.0016 to 0.0013 (see table above). This further strengthens our decision to rule out these SNPs in our analysis.

Due to these reasons, we believe that our filtering criteria are appropriate and cautious, aiming to minimize the inclusion of potential errors without significantly impacting the overall findings in the population genomics of *S. polyrhiza*. In addition, the estimated genomic diversity in American population using the MAF filtered dataset is very similar to the results reported by another research group that used a different approach (Ho *et al.* 2019).

In the revision, we have revised the manuscript to include these detailed statistics to further clarify the rationale behind our filtering process (lines 352-360).

Reference:

Linck, E. and Battey, C.J., 2019. Minor allele frequency thresholds strongly affect population structure inference with genomic data sets. *Molecular Ecology Resources*, 19(3), pp.639-647.

Pfeifer, S.P., 2017. From next-generation resequencing reads to a high-quality variant data set. *Heredity*, 118(2), pp.111-124.

Ho, E.K., Bartkowska, M., Wright, S.I. and Agrawal, A.F., 2019. Population genomics of the facultatively asexual duckweed *Spirodela polyrhiza*. *New Phytologist*, 224(3), pp.1361-1371.

2. Additionally, the requirement for SNPs to be biallelic is not adequately justified in the paper. Biallelic filters can restrict the scope of genetic diversity by excluding multiallelic variants, which may be informative for understanding the plant's genomic landscape. An explanation for why only biallelic SNPs were considered would strengthen the study's methodology.

As we replied in the previous question, in the original VCF file, the percentage of multiallelic SNPs corresponds to only 5.36% of all SNPs. The inclusion of multiallelic variants has merely no impact on our main conclusion of the low genomic diversity in *S. polyrhiza*.

Because many commonly used statistical methods and software in population genomics are optimized for biallelic SNPs and the primary focus of our study was to assess the overall

level of genomic diversity in *S. polyrhiza*, we think biallelic SNPs should provide a reliable and sufficient measure.

3. Ln. 121 : The focus on MADS-box genes for the association with structural variations (SVs) is narrow for a genome-wide analysis. While these genes may be relevant, exclusively concentrating on them risks overlooking the role of other gene families in explaining the asexuality of *Spirodela polyrhiza*. A rationale for why MADS-box genes were specifically selected for this analysis should be provided.

Thanks for this suggestion. We now tested the distribution and enrichment of small INDELS and SVs among all gene families. We found that among 118 gene families (identified based on *Arabidopsis* gene family database) that were involved in the test, 10 gene families showed significantly higher overlap fraction (enrichment) of both small INDELS and SVs compared to the overlap by chance, and the MADS-box gene family is one of them. We now included details on all these gene families in the main text (lines 118-127) and Supplementary Data 5.

Our choice to focus on MADS-box genes was intentional for several reasons: 1) MADS-box genes are well studied in plant sexual reproduction and flowering across many species, including *Arabidopsis* and *S. polyrhiza*, making them a logical starting point. 2) Among many gene families critical to flowering and sexual reproduction (e.g., MADS-box, WRKY, MYB, bZIP), MADS-box genes have the most wide existing knowledge base, enhancing our analysis and interpretations. 3) Specifically in *S. polyrhiza*, MADS-box genes are among the few gene families (probably the only one) that have been extensively studied in relation to sexual reproduction (Gramzow, L. and Theißen, G., 2020; Yoshida, A *et al.* 2021)

In summary, these factors collectively make MADS-box genes a logical and promising starting point for our analysis. However, we also provided the comprehensive analysis for all gene families in the results.

Reference:

Gramzow, L. and Theißen, G., 2020. Stranger than Fiction: Loss of MADS-Box Genes During Evolutionary Miniaturization of the Duckweed Body Plan: Loss of MADS-Box Genes in Duckweeds. *The Duckweed Genomes*, pp.91-101.

Yoshida, A., Taoka, K.I., Hosaka, A., Tanaka, K., Kobayashi, H., Muranaka, T., Toyooka, K., Oyama, T. and Tsuji, H., 2021. Characterization of frond and flower development and identification of FT and FD genes from duckweed *Lemna aequinoctialis* Nd. *Frontiers in Plant Science*, 12, p.697206.

4. LN. 156: Before concluding on the recent species split, the authors should consider confounding factors like population size, migration, and genetic drift that could influence both the nucleotide diversity (π) and the non-synonymous to synonymous substitution ratio (π_N/π_S). These factors can confound the interpretation of genomic data.

Agreed. We modified this sentence accordingly (line 160-163).

5. LN 162: While the paper attributes variations in linkage disequilibrium (LD) and recombination rates among populations to differing frequencies of sexual reproduction, it's important to also consider the impact of mutation rates on these metrics. Mutation rate is

another key factor that can influence both LD and recombination rates and should be accounted for in the analysis.

We agree with the reviewer that the mutation rate can influence both LD and recombination. Recombination can be favorably selected if it breaks negatively associated loci. Mutation can have a similar impact because it can also break LD blocks on which selection can act, thus indirectly influencing recombination. However, according to Barton (2010) this effect is visible only if the mutation rate is very large, and even so it only explains high recombination rates in a few species and it's not a general phenomenon (Barton, 2010). Moreover, for the particular case of *Spirodela polyrhiza* it has been shown that the mutation rate is extremely low, up to seven times lower than in other multicellular eukaryotes (Xu *et al.*, 2019, Sandler *et al.* 2020). For this reason, we believe that variation in LD and recombination rate in the present study is mostly due to differing frequencies of sexual reproduction and less so by mutation. Follow-up research that compares mutation rates among different populations in *S. polyrhiza* could add more information to this topic, but here it goes beyond the scope of our study.

6. LN 174 : The repeated focus on MADS-box genes for their association with changes in genomic diversity and levels of heterozygosity is limiting. A genome-wide or gene-family-wide analysis would offer a more comprehensive view and could potentially identify other significant genes involved in sexual reproduction. Relying solely on candidate gene analysis like MADS-box genes restricts the scope and may miss other relevant factors. This section can be merged with the last section of "Footprint of selection on the genome" and explain why MADS-box is important.

Thank you for your feedback and for pinpointing the potential limitations of focusing extensively on MADS-box genes in our analysis. As highlighted in our previous response, our emphasis on MADS-box genes is deeply rooted in their established roles in plant sexual reproduction, both in model species and *S. polyrhiza*. The substantial existing knowledge on MADS-box genes serves to underpin our analysis and offers a structured starting point for our investigation into the genetic basis of asexuality.

Following the suggestion from the reviewer, we performed a GWA using individuals' heterozygosity as the phenotype. However, we could not find any significant candidates. After examining the results in details (including the MADS-box genes), we reasoned that GWA approach is not suitable for purpose. If any genetic change causes a switch to asexual reproduction, it will lead to immediate reproductive isolation and a new population. In GWA, such causal genetic changes will be completely confounded with population structure and removed by the algorithm. In addition, as different genes might contribute to asexual reproduction in different populations (as we observed in MADS-box genes), we will have limited power to detect causal genetic changes for each population.

However, we acknowledge the limitation of our current strategy in focusing on mainly MADS-box genes and might miss many other potential genes. We now included this in our discussions.

We've thoroughly considered your suggestion concerning merging MADS-box section into the footprint of selection section. However, as our rationale of analyzing MADS-box is to identify the molecular mechanisms that are associated with changes in sexual reproduction, which can be but are not necessarily under selection. Therefore, we prefer to retain the

association between SVs and MADS-box gene result in its current placement, rather than merging it with the selection section.

7. The epigenomic analysis presented in the study is somewhat cursory. For a more thorough understanding of gene variation selection history and methylation patterns, it would be beneficial to integrate this analysis with a genome-wide footprint section. This would allow for a more comprehensive examination of how methylation patterns affect gene bodies or upstream regions with selection pressures, thereby providing a richer context for interpreting the data.

We agree with the reviewer that integrating the epigenomic with the selection analysis could bring a more comprehensive examination of selection potentially affecting methylation patterns. For this, we have now performed a “differentially methylated regions” (DMR) analysis on CpG methylation, since this is the methylation context that is known to have an impact on gene expression. From this DMR analysis we found a total of 218 genes that are differentially methylated between pairs of populations. Out of these 218 genes 25 are also under selection in specific populations according to our 3P-CLR results. We provide now a new table highlighting these genes and their functions (Supplementary Data 11). The functions of these genes, according to their Arabidopsis orthologs, range between embryo development, seed germination, root growth, metabolism of auxin precursors, and others. We have now added this information in the Supplementary Information section 1.6; (methods see Additional Methods section 2.7).

8. Lastly, it is imperative that all NGS raw reads be uploaded to a public repository and made accessible to reviewers. This is crucial for the verification and reproducibility of the study's findings.

All the raw NGS data stored in NCBI is now fully accessible (under the NCBI Accession number PRJNA701543).

Reviewer #4 (Remarks to the Author):

Wang et al. conduct a population scale study of genetic and DNA methylation variation in duckweed indicating that the history of asexual reproduction in this species has had genome-wide genetic and epigenomic effects. This is fitting with other studies by this group and others. Genome scans and analysis of genetic variants identifies a number of interesting candidate mutations and genes that affect sexual reproduction and that may underly reproductive differences in these populations. My major issue with this paper is that the authors put forward a number of very strong claims often on weak evidence. More care should be taken to present results in accordance to the level of evidence at hand.

1. Line 66: “epigenetic markers”, recommend changing to “chromatin marks”

We have now changed “epigenetic markers” to “chromatin marks”.

2. Line 66: use “e.g.” instead of “i.e.” as there are other chromatin features besides DNA methylation

We are now using “e.g.” instead of “i.e.”.

3. Line 66 “DNA methylations” should be “DNA methylation”
We have now changed “DNA methylations” to “DNA methylation”.

4. Lines 69-70: As currently written, it makes it sound as if CpG and CHG can be maintained by either MET1 or CMT3 interchangeably. I recommend modifying this sentence to “CpG and CHG methylation are maintained by methyltransferases1 (MET1) and CHROMOMETHYLASE3 (CMT3), respectively,”
We have now changed this sentence as you suggested. (line 68-69)

5. Lines 70-71: RdDM is capable of methylating all sequence contexts, not just CHH (see Matzke & Mosher 2014 Nature Reviews Genetics).
Thank you for pointing out this issue. We have changed this as follows: “In plants, cytosine methylation can occur in three sequence contexts: CpG, CHG, and CHH (H = A, T, or C), which are controlled by different mechanisms and have different dynamics during reproduction. Typically, CpG and CHG methylation are maintained by methyltransferases1 (MET1) and CHROMOMETHYLASE3 (CMT3), respectively. Whereas CHH methylation is mostly maintained by CMT2”. (lines 66-70)

6. Lines 71-72: The cited paper does not support a role of CMT3 in maintaining CHH methylation. Rather the authors found hypermethylation of CHH at the Onsen transposon due to compensation by CMT2 in *cmt3* mutants.
Sorry for this mistake. We have removed the sentence and the citation because this information is neither precise nor relevant to our main topic.

7. Line 101: “genomic variations” should be “genomic variation”
We have now changed “genomic variations” to “genomic variation”.

8. Line 110-112: The authors claim that the genome-wide nucleotide diversity at neutral sites is the lowest amongst multicellular organisms. However, they provide no citations to support this nor do they provide any data from other species that would demonstrate this. The evidence provided by the authors for this claim is insufficient.

We thank the reviewer for bringing this up. We have now added nucleotide diversity information for many model organisms including wild and domesticated species and corroborated that the nucleotide diversity in *S. polyrhiza* falls within the lower range. We added the new results in the Supplementary Table 2 and Supplementary Fig. 2 as well as in the main text along lines 108-113. We updated our claim that *S. polyrhiza* has very low nucleotide diversity.

9. Line 118-119 & 290-291: The authors argue that the “abundance of the SVs is substantially lower than all reported multicellular organisms.” They only cite two papers for this claim. The first is for grapevine and the second is for peach. Structural variation has been studied in many multicellular organisms and the authors do not provide sufficient evidence for their claim. Grapevine in particular is a not a good choice for comparison as it is

a highly heterozygous species that has accumulated extensive deleterious mutations due to its long history of clonal propagation.

Thank you very much for pointing out this unprecise statement. We have removed the sentence "The abundance of the SVs is substantially lower than all reported multicellular organisms." since we don't have enough data to support this claim and this observation is not relevant to the main topic of our manuscript.

10. Line 120-122: The authors claim the MADS box proteins have a higher overlap with SVs. However, the reported p-value is 0.063 which would be considered not-significant by most standards.

We agree with this feedback. However, following the suggestion from another reviewer, this whole section has been re-analyzed and updated. More specifically, we tested the distribution and enrichment of small INDELS and SVs among all gene families, and not only for MADS-box genes. We found that among 118 gene families that were involved in the test, 10 gene families showed significantly higher overlap fraction (enrichment) of both small INDELS and SVs compared to the overlap by chance, and the MADS-box gene family is one of them (enrichment for small INDELS $p=0.0006$ and for SVs $p=0.002$). We now included details on all these gene families in the main text (lines 118-127) and Supplementary Data 5.

11. Line 124-125: The authors argue that the SVs indicate either positive or relaxed selection on the MADS box proteins. I do not find the evidence convincing. First, the first result of overlap with SVs is not significant. Secondly, we are given no details as to the potential effects of these variants. Supplementary Table 2 appears to lump the SVs with the SNPs and includes upstream & downstream regions. Its impossible to ascertain what the actual impact of the SVs are on genes, let alone on the MADS box genes.

Thanks for bringing our attention to this. Concerning the first point, as explained in the previous comment, we have now statistical evidence of the significant overlap between MADS-box genes and SVs. Concerning the second point, we have provided an additional table (Supplementary Data 3) listing all the annotated information for the SVs, as well as their positions relative to (adjacent or overlapping) genes. This should offer a clearer view of the potential impact of these SVs on the annotated genes of *S. polyrhiza*.

For the third point, we apologize for the unclear phrasing. We agree that it is crucial to discuss the potential impact of SVs on genes, especially in comparison to SNPs. Unlike SNPs, SVs can have more readily inferable impacts on genes. For instance, a sufficiently large deletion that spans an entire gene would apparently result in the gene's loss of function. Similarly, insertions or deletions that overlap with coding regions can easily alter the open reading frames, potentially leading to changes in the genes' expression and function. Given the critical role of MADS-box genes in plant sexual reproduction and organ development, it is plausible that variations affecting these genes could directly impact the fitness of the organism. Therefore, the observation of the genomic variations enriched in MADS-box genes might be a hint of relaxed purifying selection or directional selection during evolution. In addition to that, our selection analysis based on the SNP data also found that five MADS-box genes have been under selection (which constitutes a significant enrichment) in the India population. This direct evidence further strengthened our inference.

However, we agree with the reviewer that we don't have direct evidence of small INDELS or SVs are indicative of selection, we thus removed our phrase in our main text.

12. Line 180-182: The authors state: "The insertion likely disrupted the function of the AGL62-like gene, thus reducing the suppression of endosperm development, which is likely required for sexual reproduction (Fig. 2a)." This is certainly an interesting hypothesis, but it is also speculation and no experimental data is given to show that AGL62 has the same function in duckweed or that this is indeed the effect of the variant. I would highly recommend that the authors tone down the language and instead offer it as a hypothesis. Thank you for pointing this out. We have changed the sentence as follows: "The insertion may potentially disrupt the function of the AGL62-like gene, suggesting a possible reduction in the suppression of endosperm development, which might be required for sexual reproduction (Fig. 2a)." (lines 182-184)

13. Line 205: "epigenomic variations" should be "epigenomic variation"
We have now changed "epigenomic variations" to "epigenomic variation".

14. Line 249 & 251. The authors claim that CMT3 is involved with maintaining CpG methylation. I do not understand their argument. No functional data is given to demonstrate that CMT3 is maintaining CpG. Different chromatin pathways interact and compensate for one another, such that when one particular pathway is affected, there can be downstream effects. A good example of this is the aforementioned study on the Onsen transposon, where *cmt3* mutations result in compensatory changes in the activity of CMT2. There are also other known indirect relationships between CMT3 and CpG methylation, such as the relationship of CMT3 with CpG gene body methylation. However, there is no direct functional evidence that CMT3 is involved in the maintenance of CpG methylation. The authors should be more cautious in the claims they put forward.

Thank you very much for pointing out this issue. We have now revised this paragraph as follows: "In the SE-Asian population, we found that gene SpGA2022_051517, a CHROMOMETHYLASE3 (CMT3) homolog in *A. thaliana*, is likely associated with maintaining CHG methylation. This is consistent with the higher CHG methylation level observed in the SE-Asian population when compared to the European and Indian populations." (lines 259-263)

Reviewers' comments:

Reviewer #1 (Remarks to the Author):

The authors have answered my questions well. I think it is valuable for publication on Communications biology.

Reviewer #2 (Remarks to the Author):

I had serious reservations about the first version of this paper when I reviewed it. I continue to have many of the same concerns, though I list only three specific examples below. As I said in my earlier review, the data are valuable and interesting and deserve to be published. I struggle with my recommendation on this paper. The interpretations are clearly not interpretations I would put forward if I were reporting on these data. They may be, however, interpretations that other competent investigators would put forward. The authors have tempered their statements about the parts of this work that, to me, represent the most significant overreach. On balance, I recommend this paper for publication, but I am not enthusiastic about my recommendation.

Abstract - Genes involved in flowering and embryogenesis might be under positive selection for many reasons, including selection for increased levels of sexual reproduction. I find it misleading to write the abstract in a way that makes it sound as if positive selection on these genes is evidence for a role of asexuality.

II. 103-105: To me a dN/dS ratio of 0.37 still indicates a lot of purifying selection. It means that, on average, 60 percent of non-synonymous mutations are eliminated by natural selection.

II. 152-162: I remain concerned about the LD/recombination analyses. First, the estimate of recombination rate is inextricably linked to the estimate of LD. They aren't independent estimates. They are different ways of describing the same phenomenon. Second, LD often reflects internal population structure rather than physical linkage. The authors' response to Reviewer #2 is insufficient. That fastSTRUCTURE fails to reveal structure doesn't mean that it's not there. Several analyses have shown that the UK-Biobank harbors hidden genetic structure that has a significant influence on inference even after attempts to statistically remove it, and that's in a highly sexual species, i.e., humans.

Reviewer #3 (Remarks to the Author):

SNP part

- Questions for the author's responses
- The SNP filtering process includes removing clusters of SNPs where there are two or more SNPs

within a 10 bp window, as described in the Materials and Methods. Based on my experience, finding two SNPs within 10 bp is relatively common and can occur by mutation, not necessarily indicating a true SNP cluster. For instance, this criterion may erroneously eliminate two closely spaced SNPs within a 1kb region that do not constitute a genuine cluster. A more appropriate threshold might be, for example, 200 SNPs within a 1kb region to define a cluster.

- Furthermore, even if some SNP clusters are genuine, a resulting SNP cluster rate of 44.51% seems excessively high. This raises concerns about the mapping procedure and the integrity of the reference genome used. It may be beneficial to review the alignment parameters and the quality of the reference sequence to ensure accurate SNP identification.

- A quick literature review suggests that SNP clusters should not be dismissed outright as false positives; they can also signify rapidly evolving genomic regions that are potentially interacting with environmental factors such as disease [1]. It is important to analyze these regions further to understand their biological relevance and the evolutionary pressures.

- From my perspective, duckweed, which reproduces through vegetative means, would be expected to form populations of clones with relatively low genetic diversity. Nevertheless, it is reasonable to anticipate the existence of sufficient SNPs or mutations that enable adaptation to the dynamic environment. Even with clonal propagation, mutations occurring during DNA replication or due to external stressors can introduce genetic variability that may be subject to natural selection, thus contributing to the adaptability of the population.

- Therefore, I suggest that the SNP cluster filtering criterion be either omitted or the window size be broadened to a kilobase scale. This adjustment may more accurately reflect the SNP number present within vegetatively propagated populations like duckweed.

- After revising the SNP filtering approach, it would be recommended to re-perform subsequent analyses, including selection analysis, linkage disequilibrium (LD) and recombination assessments, to ensure the accuracy and integrity of the results.

[1]: Long X, Xue H. Genetic-variant hotspots and hotspot clusters in the human genome facilitating adaptation while increasing instability. *Hum Genomics*. 2021;15:19.

Methylation part

- Questions for the author's responses

- Before discussing the genes involved in the maintenance of methylation, the following points should be addressed to ensure the reliability.

- The authors have conducted comparative analysis of genome-wide methylation using bisulfite sequencing data. The provided figure depict discrete differences in methylation patterns, specifically showing top methylation levels at CpG sites and least methylation level at CHH sites within gene bodies for the American group. However, the methylation level at CHH sites appears conspicuously low compared to other populations, raising concerns about the accuracy of the methylation analysis procedure.

- In the Methods and Materials section, it is mentioned that bisulfite-sequencing reads were aligned to a reference genome. However, different accessions of *Spirodela* may exhibit variations in CpG context due to the presence of SNPs or structural variations. These genetic variations can alter CpG sites and, as a result, affect bisulfite conversion patterns and the interpretation of methylation data. It should be clarified how these variations accounted for during analysis and should be presented the variation status especially for the CpG, CHG, CHH contexts and also for the non-cytosine context in reference genome that become cytosine context in other accessions.

Reviewers' comments:

Reviewer #1 (Remarks to the Author):

The authors have answered my questions well. I think it is valuable for publication on Communications biology.

Thank you.

Reviewer #2 (Remarks to the Author):

I had serious reservations about the first version of this paper when I reviewed it. I continue to have many of the same concerns, though I list only three specific examples below. As I said in my earlier review, the data are valuable and interesting and deserve to be published. I struggle with my recommendation on this paper. The interpretations are clearly not interpretations I would put forward if I were reporting on these data. They may be, however, interpretations that other competent investigators would put forward. The authors have tempered their statements about the parts of this work that, to me, represent the most significant overreach. On balance, I recommend this paper for publication, but I am not enthusiastic about my recommendation.

Thank you for your evaluation. We now rephrased the conclusions to avoid overreaching our statements. See detailed responses below.

Abstract - Genes involved in flowering and embryogenesis might be under positive selection for many reasons, including selection for increased levels of sexual reproduction. I find it misleading to write the abstract in a way that makes it sound as if positive selection on these genes is evidence for a role of asexuality.

We now rephrased the referred sentence and toned down the conclusions. In addition, to reduce the abstract to 150 words, we rephrase the whole abstract. It now reads as "Many plants are facultatively asexual, balancing short-term benefits with long-term costs of asexuality. During range expansion, natural selection likely influences the genetic controls of asexuality in these organisms. However, evidence of natural selection driving asexuality is limited, and the evolutionary consequences of asexuality on the genomic and epigenomic diversity remain controversial. We analyzed population genomes and epigenomes of *Spirodela polyrhiza*, (L.) Schleid., a facultatively asexual plant that flowers rarely, revealing remarkably low genomic diversity and DNA methylation levels. Within species, demographic history and the frequency of asexual reproduction jointly determined intra-specific variations of genomic diversity and DNA methylation levels. Genome-wide scans revealed that genes associated with stress adaptations, flowering and embryogenesis were under positive selection. These data are consistent with our hypothesis that natural selection can shape the evolution of asexuality during habitat expansions, which alters genomic and epigenomic diversity levels."

II. 103-105: To me a dN/dS ratio of 0.37 still indicates a lot of purifying selection. It means that, on average, 60 percent of non-synonymous mutations are eliminated by natural selection.

We agree, the purifying selection is indeed still strong. We referred this as “relatively weak” purifying selection because it is the highest among all species with known π_N/π_S ratio (<https://doi.org/10.1093/molbev/msx088>). We now changed the wording to “relatively relaxed purifying selection”.

The sentence now reads as “The species-wide efficacy of selection (π_N/π_S ratio) is 0.37, the highest among studied organisms(42), indicating a relatively relaxed purifying selection in *S. polyrhiza*, despite its large effective population size(36, 37).”
See line 101 in the revised version.

II. 152-162: I remain concerned about the LD/recombination analyses. First, the estimate of recombination rate is inextricably linked to the estimate of LD. They aren't independent estimates. They are different ways of describing the same phenomenon. Second, LD often reflects internal population structure rather than physical linkage. The authors' response to Reviewer #2 is insufficient. That fastSTRUCTURE fails to reveal structure doesn't mean that it's not there. Several analyses have shown that the UK-Biobank harbors hidden genetic structure that has a significant influence on inference even after attempts to statistically remove it, and that's in a highly sexual species, i.e., humans.

We agree that LD and recombination rate are intrinsically linked. We now rephrased the wordings and avoided the impression that LD and recombination rate are two independent measures.

Now, it reads as “The Indian and American populations had intermediate LD decay. Consistently, the Asian population had the highest recombination rate compared to the other three (Fig. 2b).”

See line 154 in the revised version.

Reviewer #3 (Remarks to the Author):

SNP part

- Questions for the author's responses
- The SNP filtering process includes removing clusters of SNPs where there are two or more SNPs within a 10 bp window, as described in the Materials and Methods. Based on my experience, finding two SNPs within 10 bp is relatively common and can occur by mutation, not necessarily indicating a true SNP cluster. For instance, this criterion may

erroneously eliminate two closely spaced SNPs within a 1kb region that do not constitute a genuine cluster. A more appropriate threshold might be, for example, 200 SNPs within a 1kb region to define a cluster.

Thank you for raising this point. In our filtering, we considered SNP clusters as more than **three** SNPs within 10bp, which is a common filtering used in different studies and tutorials (e.g., <https://southgreenplatform.github.io/trainings//files/TPmapping.pdf> ; <https://gatk.broadinstitute.org/hc/en-us/community/posts/20737428876571-About-RNA-seq-snp-filtration>; <https://bmcgenomics.biomedcentral.com/articles/10.1186/1471-2164-11-469>).

In our previous work (<https://www.nature.com/articles/s41467-019-09235-5>), we have evaluated different SNP filtering parameters and evaluated the identified SNPs with Sanger-sequencing. Our results suggested that many SNPs that clustered together were false-positives and using our current filtering parameters led to the most accurate SNP detection without reducing false negatives (e.g., via simulations).

In addition, using our filtering parameters, all of the estimated genomic diversity and LD parameters are very similar to another study that carried out by another research group (<https://nph.onlinelibrary.wiley.com/doi/full/10.1111/nph.16056>), which used another SNP filtering strategy to remove non-reliable SNPs.

In addition, we followed the suggestions from the reviewer and performed SNP filtering steps using the suggested parameters (e.g., 200 SNPs in 1kb region). This resulted in slightly higher number of SNPs (18.7%). We had a close look at these 18.7% SNPs and found that most of them (>88%) overlapped with transposable elements and SVs or being adjacent to small INDELs (see detailed explanation below), suggesting these SNP clusters are likely due to misalignment.

We further performed downstream analysis using the new SNP cluster filtering parameters (200SNPs in 1kb). We found that the conclusion on genomic selection and low genomic diversity remained very similar to our original analyses, suggesting that our conclusions are robust and largely independent of SNP filtering parameters.

In the revision, we decided to keep our original SNP cluster filtering strategy for the following reasons: 1) our previous study (<https://www.nature.com/articles/s41467-019-09235-5>) and other publications suggested the current filtration of SNP cluster (three or more SNPs within 10 bp window) is justified. 2) A broadened SNP cluster filtration might keep a small number of true SNPs but also introduced inevitable artifacts through misalignment against the complex genomic regions like TEs, SVs etc. 3) The results of downstream analyses based on the broadened SNP cluster filtration remained high similarity to our current findings, and our main conclusion remains unchanged.

However, for the transparency, we summarized the results from using the new SNP cluster filtering strategy (200SNPs in 1kb) in the method section.

It now reads as " To exam whether SNP cluster filtering criterion affects the estimation of genomic diversity and selection, we performed additional analyses based on a more relaxed filtering parameter (≥ 200 SNPs in 1 kb region). Although the second SNP cluster filtering criterion resulted in 18.7% more SNPs, which are mostly

(>88%) located in TE regions or nearby the SV or INDELS, the patterns of genomic diversity and selection did not change.”

See line 329-334 in the revision.

- Furthermore, even if some SNP clusters are genuine, a resulting SNP cluster rate of 44.51% seems excessively high. This raises concerns about the mapping procedure and the integrity of the reference genome used. It may be beneficial to review the alignment parameters and the quality of the reference sequence to ensure accurate SNP identification.

The 44.51% rate seems high. However, many of the SNP clusters had very low minor allele frequency (less than 1%), largely due to the mapping error from a single individual. Even we don't filter the SNP clusters, many of these SNPs will also be removed after filtering low minor allele frequency step.

The reference genome we used is of high quality and the mapping rates are very high. The same mapping parameters and reference genome were used in our previous work (<https://www.nature.com/articles/s41467-019-09235-5>). In that study, we have validated the quality of SNP callings using Sanger-sequencing as well as simulations. In addition, the results were also independently confirmed by another study (<https://www.ncbi.nlm.nih.gov/pmc/articles/PMC7642947/>).

Taken together, we believe that the reference genome and mapping parameters are justified.

- A quick literature review suggests that SNP clusters should not be dismissed outright as false positives; they can also signify rapidly evolving genomic regions that are potentially interacting with environmental factors such as disease [1]. It is important to analyze these regions further to understand their biological relevance and the evolutionary pressures.

We do agree that SNP clusters can be important and biologically relevant. In our previous study (<https://www.nature.com/articles/s41467-019-09235-5>), we initially did not remove SNP clusters and identified many putatively SNPs. However, after many Sanger-sequencing efforts, we noticed that many of them are false positives, especially when they are at a low allele frequency from the population study. After considering the high false positive rates and relatively low false negative rates, we decided to use with the original SNP cluster filtering criterion (see details above).

- From my perspective, duckweed, which reproduces through vegetative means, would be expected to form populations of clones with relatively low genetic diversity. Nevertheless, it is reasonable to anticipate the existence of sufficient SNPs or mutations that enable adaptation to the dynamic environment. Even with clonal propagation, mutations occurring during DNA replication or due to external stressors can introduce genetic variability that may be subject to natural selection, thus contributing to the adaptability of the population.

We agree. Our results suggest that genetic diversity in *S. polyrhiza* is relatively low, but

still substantial. For a comparative perspective, the levels of genetic diversity in *S. polyrhiza* are similar to the diversity found in *Arabidopsis thaliana* in North America.

- Therefore, I suggest that the SNP cluster filtering criterion be either omitted or the window size be broadened to a kilobase scale. This adjustment may more accurately reflect the SNP number present within vegetatively propagated populations like duckweed.

See the response mentioned above. We believe our current SNP cluster filtering criterion is appropriate. However, to make it transparent, we also summarized the results using the suggested filtering criterion in the supplementary information. The results and conclusions using these two SNP cluster filtering criteria are overall very consistent. We now added this information in the method section. See line 329-334.

- After revising the SNP filtering approach, it would be recommended to re-perform subsequent analyses, including selection analysis, linkage disequilibrium (LD) and recombination assessments, to ensure the accuracy and integrity of the results. We followed the suggestion and re-performed the subsequent analyses. Based on this broadened filtration, the estimated genome-wide π is 0.002, which is still very low compared to other natural populations. LD decay analysis remained unchanged (Response Figure 1). As for the selection analysis, taken LASSI analysis as an example, we found the selective signals across the genomes highly consistent with previous analysis (Response Figure 2), and around 86% of under-selection genes are the same compared to the previous run of LASSI. Together, the results suggest that our conclusion is independent of SNP filtering parameters.

Response Figure 1. LD decay analysis across four populations and based on the SNP dataset generated from the broadened SNP cluster filtration. The LD decay patterns of all populations are highly consistent with that from previous analysis.

Response Figure 2. Comparison of two runs of LASSI selective scan. “T previous” indicates the T statistics calculated based on previous SNP cluster filtration (three or more SNPs within 10 bp), while “T current” indicates the broadened SNP cluster filtration. Both runs found very similar selection signals that more than 86% of under-selection genes remained the same.

[1]: Long X, Xue H. Genetic-variant hotspots and hotspot clusters in the human genome facilitating adaptation while increasing instability. Hum Genomics. 2021;15:19.

Methylation part

- Questions for the author's responses
- Before discussing the genes involved in the maintenance of methylation, the following points should be addressed to ensure the reliability.
- The authors have conducted comparative analysis of genome-wide methylation using bisulfite sequencing data. The provided figure depict discrete differences in methylation patterns, specifically showing top methylation levels at CpG sites and least methylation level at CHH sites within gene bodies for the American group. However, the methylation level at CHH sites appears conspicuously low compared to other populations, raising concerns about the accuracy of the methylation analysis procedure.

The low CHH sites in *S. polyrhiza* has been repeatedly reported in several studies (<https://academic.oup.com/g3journal/advance-article/doi/10.1093/g3journal/jkae004/7513072>; <https://onlinelibrary.wiley.com/doi/pdf/10.1111/tpj.13400>);

See text in line 277-278

- In the Methods and Materials section, it is mentioned that bisulfite-sequencing reads

were aligned to a reference genome. However, different accessions of *Spirodela* may exhibit variations in CpG context due to the presence of SNPs or structural variations. These genetic variations can alter CpG sites and, as a result, affect bisulfite conversion patterns and the interpretation of methylation data. It should be clarified how these variations accounted for during analysis and should be presented the variation status especially for the CpG, CHG, CHH contexts and also for the non-cytosine context in reference genome that become cytosine context in other accessions.

Thank you for raising up the potential impact of genetic variations on methylation analysis. To address concerns about how genetic variations might affect the interpretation of methylation data, we have redone our methylation analyses.

Following the previous studies (<https://www.nature.com/articles/nature11968>; <https://pubmed.ncbi.nlm.nih.gov/27419873/>; <https://www.nature.com/articles/s41467-020-19333-4>), we performed SNP substitution for each genotype to create pseudo-reference genomes. Subsequently, we realigned the bisulfite-sequencing (BS-seq) reads to these pseudo-references. Based on these alignments, we reanalyzed the weighted methylation levels (wML) across genomic regions for four populations. We also re-conducted the methylation clustering and identified the differentially methylated regions among populations.

Overall, we observed only minor changes when compared to previous results, but the patterns and conclusions remain the same. We observed a slight increase in the mapping rate of BS-seq reads. The wML across different genomic regions did not show significant changes. Our main conclusions remain unchanged: compared to other angiosperms, *S. polyrhiza* show exceptionally low wML in CpG, CHG, and CHH methylation, with differences in CHH wML observed among sexual and asexual populations. In the revision, we updated the figures and texts based on the new analyses.

The Fig. 3 is now updated as following:

Fig. 3: Weighted methylation level (wML) among four populations. a-c Bar plots of genome-wide weighted methylation level (wML) in (a) CpG, (b) CHG, and (c) CHH context (N=5). Black vertical lines indicate the standard error of the mean of each population's wML. Uppercase letters indicate statistical differences among populations using Wilcoxon test (with bonferroni method for multiple tests corrections). d-f wML in the gene body and its flanking 2 kb regions in (d) CpG, (e) CHG, and (f) CHH context. g-i wML in TE and its flanking 2 kb regions in (g) CpG, (h) CHG, and (i) CHH context. The asterisks indicate significant differences between populations ($P < 0.05$; Wilcoxon test), while "n.s." indicates no significant difference. See changes in line 797-806.

In addition, we found that using the suggested pipeline, the correlation between DNA methylation and genetic distances disappeared for CpG context, while the CHG and CHH context remained the same. Now revised the text in the manuscript according to the updated results. It now reads as: "The hierarchical clustering of 20 methylomes in CHG and CHH contexts in gene bodies show overall consistency with their genetic similarity (Supplementary Fig. 13 and 14) with few discrepancies were mostly found within the same population or between the recently diverged SE-Asian and European populations. While in the CpG context, we did not observe clear correlations between genetic and methylation distances (Fig. 15)." See line 190-194, and supplementary fig. 13-15.

REVIEWERS' COMMENTS:

Reviewer #3 (Remarks to the Author):

All concerns that I raised were well resolved, and I have no objection to the publication.